# Superior robustness of anomalous non-reciprocal topological edge states

Zhe Zhang[1], Pierre Delplace[2] & Romain Fleury[1✉]

Robustness against disorder and defects is a pivotal advantage of topological systems[1], manifested by the absence of electronic backscattering in the quantum-Hall[2] and spin-Hall effects[3], and by unidirectional waveguiding in their classical analogues[4,5]. Two-dimensional (2D) topological insulators[4–13], in particular, provide unprecedented opportunities in a variety of fields owing to their compact planar geometries, which are compatible with the fabrication technologies used in modern electronics and photonics. Among all 2D topological phases, Chern insulators[14–25] are currently the most reliable designs owing to the genuine backscattering immunity of their non-reciprocal edge modes, brought via time-reversal symmetry breaking. Yet such resistance to fabrication tolerances is limited to fluctuations of the same order of magnitude as their bandgap, limiting their resilience to small perturbations only. Here we investigate the robustness problem in a system where edge transmission can survive disorder levels with strengths arbitrarily larger than the bandgap—an anomalous non-reciprocal topological network. We explore the general conditions needed to obtain such an unusual effect in systems made of unitary three-port non-reciprocal scatterers connected by phase links, and establish the superior robustness of anomalous edge transmission modes over Chern ones to phase-link disorder of arbitrarily large values. We confirm experimentally the exceptional resilience of the anomalous phase, and demonstrate its operation in various arbitrarily shaped disordered multi-port prototypes. Our results pave the way to efficient, arbitrary planar energy transport on 2D substrates for wave devices with full protection against large fabrication flaws or imperfections.

Among the unique and counter-intuitive attributes of topological systems, topological robustness[1] against disorder and flaws is undoubtedly one of the most remarkable. This property shows substantial application potential by relaxing the tight constraints imparted by fabrication tolerances, and provides a way to route energy and information in a wide variety of 2D platforms[4–27], ranging from quantum electronics[23] to classical photonic[4,5] and phononic devices[25–27]. Topological edge states were found in systems with broken time-reversal symmetry, such as Chern insulators[14,28], and then extended to time-reversal invariant scenarios, including the Z2 (ref. [3]) and other symmetry-protected schemes[29], simultaneously stimulating study of their classical analogues[6,10,17]. So far, Chern topological edge modes[14–25] undeniably represent the most reliable solution for point-to-point energy guiding, as they provide truly unidirectional, backscattering-immune wave transport at their boundaries. They have been reported in non-reciprocal artificial wave media, such as externally biased magneto-photonic crystals[16] or mechanical systems[13] with moving[17,19,20,25] or time-dependent[8,24] elements. Albeit protected from the presence of local defects by the Chern number, the edge modes cannot survive the presence of distributed disorder of sufficiently large magnitude[1,4,5,14], especially when the average amplitude of frequency fluctuations gets larger than the bandgap size. This behaviour inherently confines the topological protection of Chern phases to small distributed disorder levels.

Here we demonstrate an anomalous non-reciprocal topological phase in which edge transmission is quantitatively stronger than for the Chern phase, surviving parametric fluctuations arbitrarily larger than the bandgap size. We find such anomalous robustness in unitary scattering networks made of interconnected non-reciprocal resonant scatterers coupled by non-directed phase links. We compare quantitatively the robustness of transmission through the anomalous and Chern channels to phase-link and scattering disorder, by statistical averaging over many disorder realizations. Our experiments at microwave frequencies confirm the superior resilience of the anomalous transmission channel over the Chern one. We apply our findings to the design of ideally robust networks with arbitrarily located ports and irregular shapes, including a perfect six-port circulator.

## A non-reciprocal scattering network

Consider the non-reciprocal unitary scattering network of Fig. 1a, which consists of general three-port non-reciprocal scatterers connected by bidirectional links in a honeycomb periodic structure. The scattering elements exhibit threefold ($C_3$) rotational symmetry, while the links impart a phase delay of $\varphi$, as represented in the zoomed-in view of the unit cell (Fig. 1b). The scattering process is described by a unitary $3 \times 3$ asymmetric scattering matrix $S_0$ whose general

[1]Laboratory of Wave Engineering, School of Electrical Engineering, EPFL, Lausanne, Switzerland. [2]Univ Lyon, ENS de Lyon, Univ Claude Bernard, CNRS, Laboratoire de Physique, Lyon, France. ✉e-mail: romain.fleury@epfl.ch

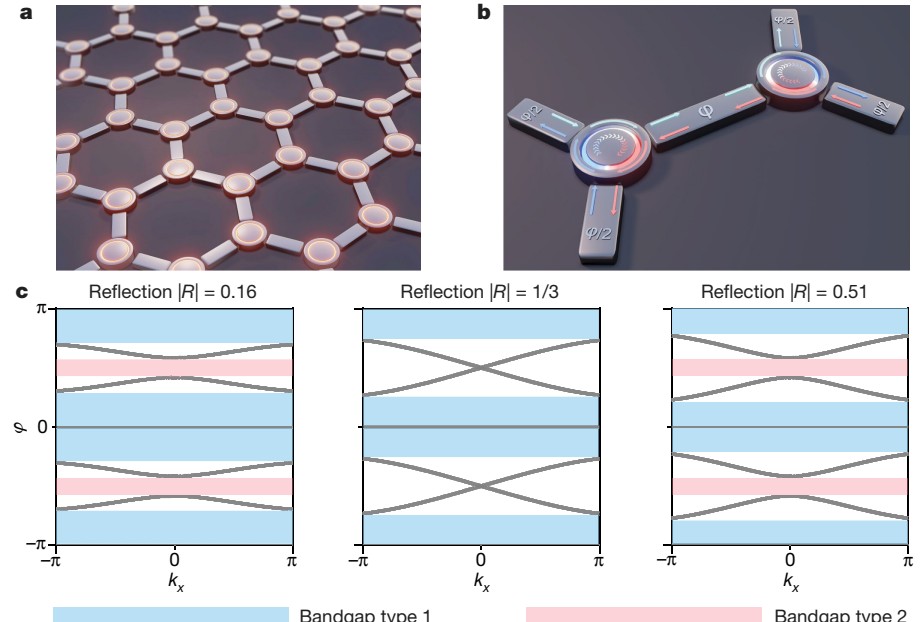

**a**

**b**

**c**

Reflection |R| = 0.16

Reflection |R| = 1/3

Reflection |R| = 0.51

Bandgap type 1     Bandgap type 2

**Fig. 1 | Topological non-reciprocal wave network and its bulk band structure. a**, We consider a unitary scattering network made of three-port non-reciprocal elements, described by asymmetric unitary scattering matrices. **b**, Unit cell of the honeycomb lattice, highlighting the signals entering the non-reciprocal elements, their 120° rotational symmetry, and the reciprocal phase delay $\varphi$ imparted by the links. The network is described by a unitary unit-cell scattering operator $S(\mathbf{k})$ defining a Floquet unitary mapping with quasi-energy $\varphi$. **c**, Evolution of the Floquet band structure on increasing the level of reflection of the non-reciprocal elements from $|R| = 0.16$ (leftmost panel, with angular parameter values $\xi = -\eta = 2.5\pi/12$) to $|R| = 0.51$ (rightmost panel, $\xi = -\eta = 3.5\pi/12$). While the type 1 bandgaps do not change much, at $|R| = 1/3$ (centre panel, $\xi = -\eta = \pi/4$), the type 2 bandgap closes, symptomatic of a topological phase transition.

parametrization involves only two angles, $\xi$ and $\eta$, in the interval ($-\pi/2$, $\pi/2$) (see Supplementary Information and Extended Data Fig. 1). The wave propagation in the infinite network can be described by a Bloch eigenproblem, which considers the scattering at the nodes, described by a 6 × 6 unitary matrix $S(\mathbf{k})$, and also involves the bidirectional phase delay $\varphi$ induced by the links:

$$S(\mathbf{k})|c(\mathbf{k})\rangle = e^{-i\varphi}|c(\mathbf{k})\rangle \tag{1}$$

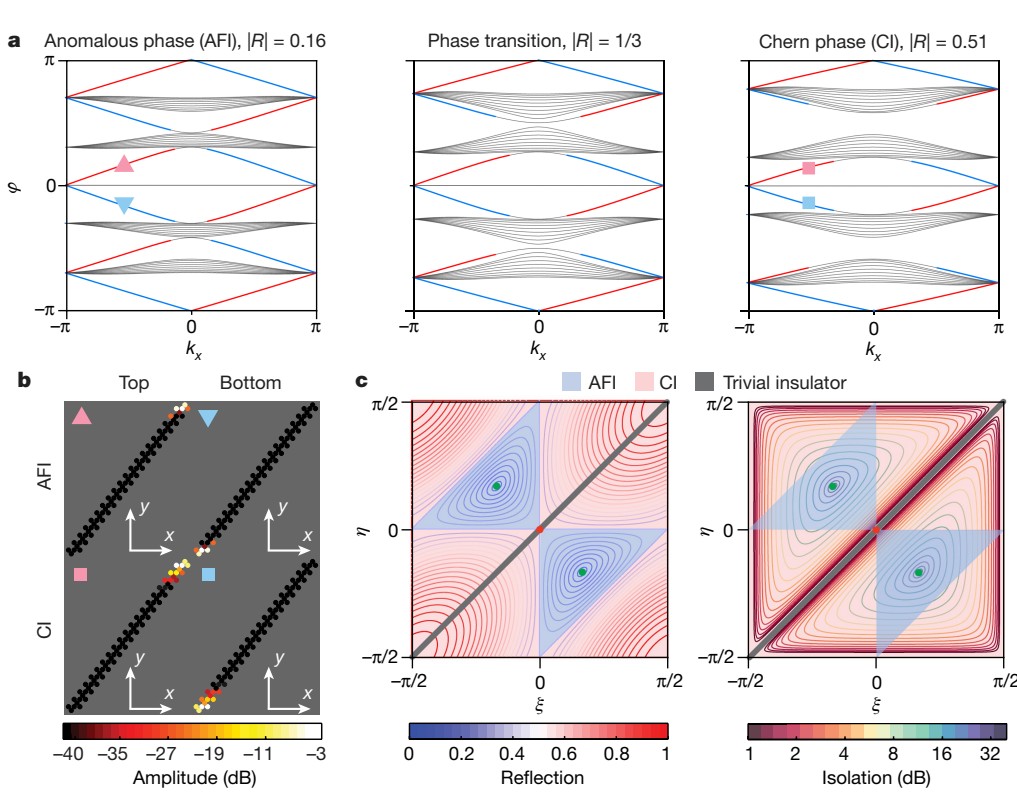

**a**  Anomalous phase (AFI), |R| = 0.16     Phase transition, |R| = 1/3     Chern phase (CI), |R| = 0.51

**b**     Top     Bottom          **c**     AFI     CI     Trivial insulator

AFI

CI

−40  −35  −27  −19  −11  −3
Amplitude (dB)

0  0.2  0.4  0.6  0.8  1
Reflection

1  2  4  8  16  32
Isolation (dB)

**Fig. 2 | Anomalous and Chern topological phases in non-reciprocal wave networks. a**, Band structure of a supercell with periodic boundary conditions along $x$ and unitary reflection at the top and bottom. The parameters are the same as in Fig. 1c. The low-reflection case is the anomalous topological phase (an anomalous Floquet insulator, AFI), which features an edge mode in every quasi-energy gap. Conversely, the high-reflection case supports edge modes only inside the type 1 bandgaps, consistent with the Chern insulator (CI) phase. Edge modes localized to the top and bottom are shown in red and blue, respectively. The phase transition is depicted in the middle panel. **b**, Supercell with examples of the profiles of Chern and anomalous topological edge modes, corresponding to the markers in **a**. **c**, Topological phase diagrams in the ($\xi$, $\eta$) plane. The blue-shaded areas correspond to the anomalous phase, and the red-shaded areas to the Chern phase. Left, comparison with the iso-reflection contours of the individual scatterers, demonstrating the coincidence between the topological phase transition and the $|R| = 1/3$ contour. Right, comparison with the non-reciprocal isolation level of the individual scatterers $|S_{21}/S_{12}|$. On the thick grey diagonals in panel **c**, the scatterers are reciprocal and the type 1 bandgaps close. At the centre red point, all bandgaps close. The two green points represents the perfect circulator cases, either with right-handed circulation (upper-left point) or left-handed circulation (lower-right point).

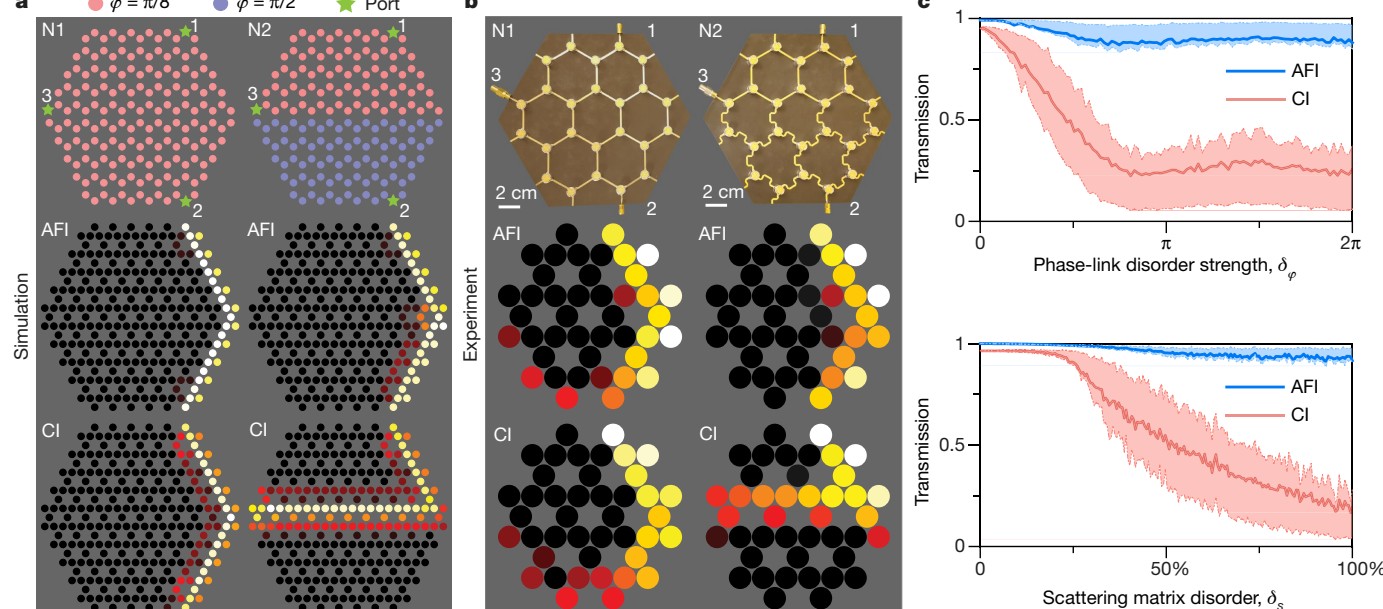

**Fig. 3 | Superior robustness of anomalous non-reciprocal topological edge transmission. a**, Numerical simulation of the steady-state energy propagation in finite non-reciprocal networks with different phase-link distributions. The signal is incident from port 1 (see top panel for positions of ports 1–3). The parameters used to generate the anomalous (centre panel) and Chern (bottom panel) phases are the same as in Figs. 1 and 2. Left column (network 1, N1), the phase-link distribution is uniform, with $\varphi = \pi/8$, and the energy can be transmitted to port 2 in both the anomalous and Chern phases. Right column (network 2, N2), we introduce an interface and abruptly change the value of $\varphi$ to $\pi/2$ for the bottom part of the network. Only the anomalous phase is robust to this change, and keeps transmitting to port 2. In the Chern phase, the edge mode travels along the interface and reaches port 3. **b**, Experimental validation using microwaves in a network made of ferrite circulators. The colourmap represents the measured field amplitude distribution, where brighter colours correspond to a large field amplitude, and darker colours a low field amplitude. **c**, Top panel, transmission between ports 1 and 2 in a disordered system with randomly generated phase delays. The phases are uniformly drawn in an interval $[-\delta_\varphi/2, \delta_\varphi/2]$ around $\varphi = \pi/8$. Solid lines represent the value of transmission averaged over 1,000 realizations of disorder, and the dashed lines are the first and last quartiles (Q1 and Q3). The anomalous edge transmission channel can survive disorder strengths up to a full $2\pi$ rotation. Bottom panel, same but for the case of scattering matrix disorder within a given topological phase ($\varphi = \pi/8$). Transmission in the anomalous channel is also more resilient to this disorder type. See Supplementary Information for particular field maps and other Chern cases.

So far, topological unitary scattering wave networks[6,30–34] have only been implemented in reciprocal systems[7,35–37] exploiting two time-reversed subspaces that are never genuinely decoupled. On the contrary, our non-reciprocal scattering network is formally analogous to a rigorously oriented kagome graph (see Supplementary Information), described by a unitary matrix[33] $S(\mathbf{k})$, which can be mapped[38] onto the Floquet eigenproblem of a periodically driven lattice[39–45], with the angle variable $\varphi$ taking the role of the quasi-energy. Therefore, we can truly benefit from both the advantages of non-reciprocity[46], and the potentially richer topological physics of Floquet systems[44].

## Chern and anomalous phases

We used the model of equation (1) to explore the parameters influencing potential topological phase transitions in the network. We found the individual reflection coefficient $|R|$ of the non-reciprocal scatterers to be the main 'control knob' for the closing of the quasi-energy bandgaps. The evolution of the bulk band structure with increasing values of $|R|$ is shown in Fig. 1c. Our semi-analytical model shows a systematic closing of two of the bandgaps at $|R| = 1/3$ (denoted type 2, in red) while the others (type 1, in blue) do not change much. This suggests that topological phase transitions may be controlled by the individual scatterer reflectance.

To confirm this intuition, we probe the existence of edge modes for each of these situations by numerically calculating the band structure of a ribbon terminated by full-reflection boundary conditions at top and bottom. As depicted in Fig. 2a, both the low- and high-reflection cases (respectively the leftmost and rightmost panels) exhibit chiral edge modes located at the walls either at the top (red line) or bottom (blue line), with profiles represented in Fig. 2b. The main difference is that the low-reflection case has edge modes in every quasi-energy bandgap, whereas at high reflection, they are

found only in bandgaps of type 1. This low-$|R|$ behaviour is the hallmark of anomalous Floquet insulators[33,35,42,45] (AFI), which possess topological edge states despite the Chern number of all surrounding bands being zero. In contrast, the high-reflection case corresponds to the Chern insulator (CI). We map out in Fig. 2c the complete topological phase diagram for every possible realization of the scattering matrix $S_0$, represented by the angle parameters $\xi$ and $\eta$. The CI and AFI regimes are shaded in red and blue, respectively. To connect this phase diagram with physically meaningful quantities, we plot it twice in the same parameter space, together with contour lines depicting the reflectance (Fig. 2c, left) and non-reciprocal isolation (Fig. 2c, right). Remarkably, the phase diagram unambiguously demonstrates the coincidence between the 1/3 reflection contours with the topological phase transition. Its centre corresponds to a semi-metallic phase, with all bandgaps closed, whereas the green point is the perfect circulator case with $|R| = 0$ and infinite isolation, for which the bulk bands are flat and the edge modes are dispersionless (see Extended Data Fig. 2). Such a critical condition corresponds to a phase rotation symmetric point[33], which can only occur in the anomalous (or trivial) phases.

## Robustness comparison

From the band structures of Fig. 2a, we can already intuitively expect the AFI edge transmission to be much more robust than the CI one to quasi-energy fluctuations, even those much larger than the bandgap size. Indeed, the AFI phase occurs in the ballistic regime, in which reflections at nodes are low, yielding relatively flat (slow) bulk bands and large bandgaps. An abrupt jump of $\varphi$ within the lattice is very likely to land in a bandgap, which necessarily carries an edge mode. Conversely, in the CI phase, the probability of an edge mode being destroyed by fluctuations larger than the bandgap is much higher, owing to the increased width of the bulk bands[33] and the

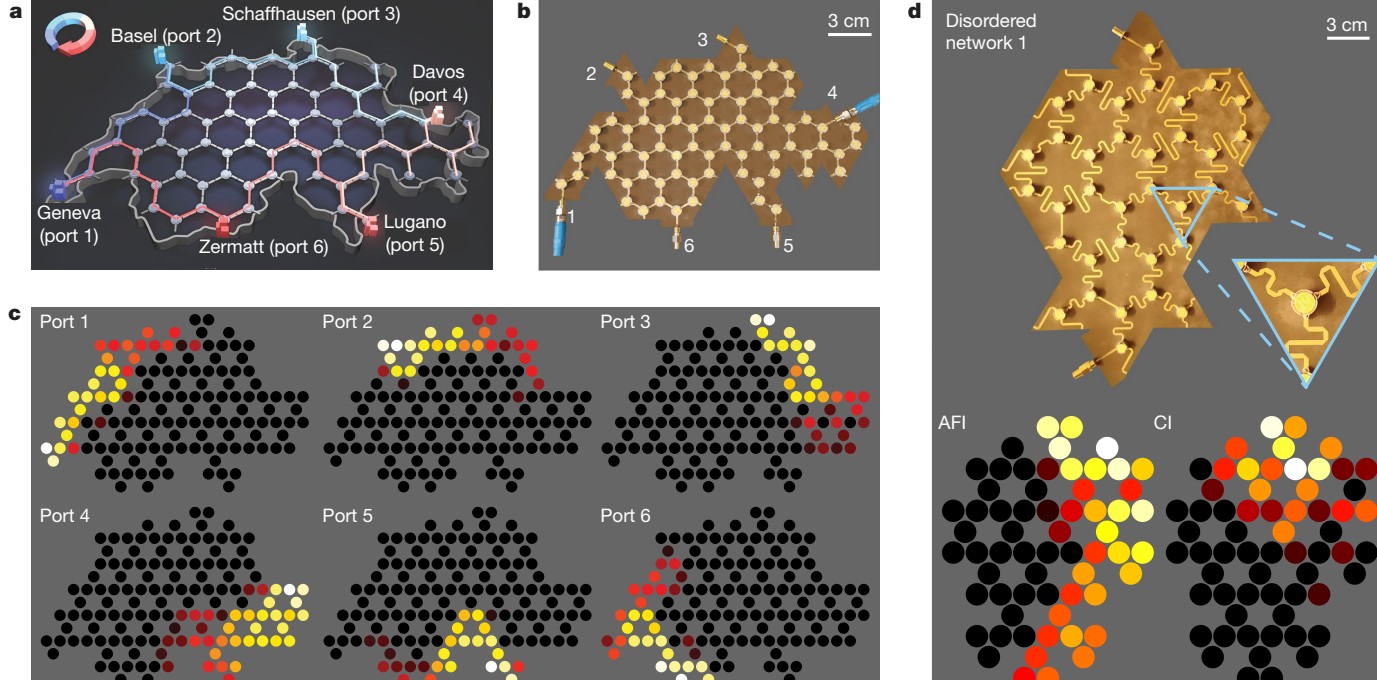

**Fig. 4 | Experiments on irregularly shaped and disordered networks. a,** We consider a network shaped like the map of Switzerland, and placed six ports on the external boundary at six city locations. **b,** Photograph of the associated prototype, showing ports 1–6. **c,** Experimental field maps upon sequential excitation of this six-port system. The network behaves as a six-port circulator despite its irregular shape, the random port locations and the high number of occurrence of trivial bandgaps.

**d,** Experimental validation of robust anomalous transmission in a two-port system with randomly disordered phase links under the largest possible disorder strength ($\delta_\varphi = 2\pi$). Top, photograph of one of our prototypes. Bottom, measured field maps in the AFI and CI cases. The AFI edge mode reaches port 2, while the Chern one is blocked. The other four results are shown in Extended Data Fig. 9.

occurrence of trivial bandgaps. As an example of such a situation, let us consider the transport properties of edge modes in a finite non-reciprocal network with an abrupt quasi-energy jump in the middle (Fig. 3a, right). As a reference, we also include the case of a uniform sample (Fig. 3a, left). The two hexagonal-shaped networks have three input/output ports, as shown in the top row of Fig. 3a. Network 1 (N1) consists of uniformly distributed phase links $\varphi = \pi/8$, while for network 2 (N2), a quasi-energy jump is introduced by changing all phase links in the bottom part to $\pi/2$. With numerical simulations, we then compare the propagation of the anomalous and Chern edge modes, when exciting from port 1. The anomalous phase finds itself in topological bandgaps at both $\varphi = \pi/8$ and $\pi/2$ (Fig. 2a, left), whereas the Chern phase possesses a nontrivial bandgap only at $\varphi = \pi/8$ (Fig. 2a, right). As shown in Fig. 3a, the anomalous edge mode crosses the interface completely unperturbed. In stark contrast, the Chern edge mode is unable to transmit to port 2 in the presence of the interface, and all the energy is guided to port 3.

We validate experimentally this fundamental distinction between the anomalous and Chern phases by designing a non-reciprocal network operating at microwave frequencies. The scatterers are ferrite circulators connected with microstrip lines. Our experimental design, which takes into account both the frequency dispersion of the scatterers and delay lines, finds itself in the anomalous and Chern phases at 4.9 GHz and 3.6 GHz, respectively. Modification of the phase delays of the links is induced by changing the total lengths of the microstrip lines with serpentine paths. As shown in Fig. 3b, the measured field amplitude profiles confirm the resilience of the anomalous edge mode to the phase jump, in perfect agreement with the numerical predictions. Further evidence is provided by the measured changes in scattering parameters and field maps upon exciting ports 2 and 3 (Extended Data Figs. 3, 4d, e, 5 and 6).

The resilience of the anomalous edge transport in these interface scenarios, involving two periodic networks, raises the question of its generalization to non-periodic quasi-energy perturbations. To answer quantitatively, we consider the same hexagonal network as in the left of Fig. 3a, and impose site-dependent disorder on the phase links, with

fluctuations of strength $\delta_\varphi$ randomly drawn with uniform probability in the interval $\pi/8 + [-\delta_\varphi/2, \delta_\varphi/2]$. We then numerically extract the transmission from ports 1 to 2 for 1,000 realizations of disorder, and plot its magnitude versus $\delta_\varphi$ in the top panel of Fig. 3c. The solid lines represent the ensemble average, and the dashed lines are the first and last quartiles (Q1 and Q3). In the clean limit ($\delta_\varphi = 0$), both AFI and CI phases show high transmission, since the edge states exist in both cases and are unperturbed. We now turn on the disorder, up to the maximal possible strength, which corresponds to randomly drawn values in the entire $2\pi$ quasi-energy range, much larger than the bandgap size of both AFI and CI phases (roughly $\pi/4$). Upon increasing $\delta_\varphi$, the average transmission in the Chern case quickly drops to low values. Remarkably, the AFI transmission shows a markedly different behaviour, remaining near 90% even when $\delta_\varphi$ reaches $2\pi$ (fully random case). Note that this exceptional robustness does not require the critical condition $|R| = 0$ to be reached, since the figure is generated for $|R| = 16\%$. Such statistically stable transmission constitutes solid evidence of the superiority of anomalous non-reciprocal topological networks, which survive phase disorder levels arbitrarily larger than their bandgap size. We also consider the other possible source of disorder, namely the scattering matrices of the nodes, which we pick randomly within the Chern or anomalous phases, fixing $\varphi = \pi/8$. The transmission statistics are shown in the bottom panel of Fig. 3c. We see that the anomalous transmission can tolerate 100% disorder in the choice of scattering matrices, whereas the Chern one falls after 25%. The reason for this surprising behaviour is that in a disordered Chern phase (random $|R| > 1/3$), transmission is mediated by both bulk and edge modes, but is blocked by trivial gaps, whereas in the anomalous case (random $|R| < 1/3$), those trivial gaps are absent (see Supplementary Fig. 8). This shows that the superior robustness of the anomalous phase is not restricted to phase-link disorder, but also to the other possible source of disorder: fluctuations of the scattering matrices.

We validate the resilience of the anomalous transmission by performing experiments on irregularly shaped disordered networks. First, we demonstrate the use of anomalous phases in a practical scenario, where an

anomalous non-reciprocal topological network is used to create a robust six-port circulator with arbitrary shape and port locations. The prototype is shaped like Switzerland, and we place six ports at the locations of six boundary cities (Fig. 4a). We aim at connecting each city to its clockwise closest neighbour, with strong non-reciprocal isolation to any other city. A picture of the fabricated prototype is shown in Fig. 4b. We sequentially excite each input of this six-port non-reciprocal network, and report the measured experimental field maps in the AFI band (Fig. 4c). Despite the presence of finite fabrication tolerances, such as the inaccuracy in the surface mounting process of the elements, and shrinking effects due to the employed reflow oven method, and regardless of the tortuous shape of the border, we observe the expected clockwise non-reciprocal circulation of the energy, consistent with simulations (Extended Data Fig. 7c). Such robustness is also observed in longer-range transmission tests between ports 1 and 4 (Extended Data Fig. 7a and b). Second, we provide an experimental validation of the superiority of the anomalous transmission in the presence of fully random phase delays. We built five different prototypes, one of them shown in Fig. 4d, with phase fluctuations in a $2\pi$ range implemented via serpentine links. The measured field maps in the AFI and CI phases show that only the anomalous channel survives such strong distributed perturbations, consistent with our statistical studies.

## Conclusion and outlook

We envision that such anomalous wave platforms may be used in a new generation of multiple-input multiple-output devices, capable of reaching an unprecedented level of robustness. Since individual reflection is the sole 'control knob' for the transition from the CI to the AFI phase, one could foresee very practical ways to reconfigure a domain wall between the two phases—for example, by simply changing the matching of the scatterers—without the need for flipping a magnetic field. Our table-top experiment, compatible with standard printed circuit board microwave technologies and off-the-shelf surface mount components, provides genuine non-reciprocity and large robustness, not only to local defects, but also to distributed imperfections. This opens an avenue to a new generation of wave systems[47] that can provide reconfigurable point-to-point unidirectional energy guiding, with arbitrary control over the imparted phase delays and full immunity against backscattering. Finally, exploration of the interplay between anomalous non-reciprocal networks and non-Hermitian perturbations (such as radiation losses occurring when coupling the edge mode to the free-space continuum) represents a promising future opportunity for topologically controlled radiation patterns in applications such as multiple beam antennas for 5G communications.

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

## Methods

### Network topology

The Chern number does not fully account for the topology of unitary operators, such as the scattering matrix in equation (1). For unitary evolutions, the eigenvalue (quasi-energy) spectrum being defined on a circle, each (quasi-energy) band is now allowed to be connected to the next one by an edge state[42]. Because of the cyclicity of the spectrum, and because the Chern number of a band counts the number of edge states that merge into that band, it follows that the Chern numbers of each band vanish. Since all the gaps are filled by a chiral edge state, this regime is called anomalous.

Actually, the topology of unitaries, such as evolution operators or our scattering matrix, is better described by the homotopy group $\pi_3(U(N)) = \mathbb{Z}$, whose elements are the topological numbers

$$W_\psi = (24\pi^2)^{-1} \int \mathrm{tr}\left(V_\psi^{-1} \mathrm{d}V_\psi\right)^3. \tag{2}$$

The power 3 must be understood in the language of differential forms, and the integral runs over a 3-torus, spanned by the quasi-momentum $\mathbf{k} = (k_x, k_y)$ and time $t$ (over a time period $T$). Time is not explicit in scattering networks. However, the cyclicity of the network makes possible a direct mapping with a Floquet (that is, $T$-periodic in time) evolution operator $U(t, \mathbf{k})$, such that an interpolation parameter that formally plays the role of time can be introduced[33]. Finally, the operator $V_\psi$ is a periodized (in time) evolution operator. For Floquet systems, it reads as[42].

$$V_\psi(t, \mathbf{k}) = U(t, \mathbf{k})\exp(itH_{\mathrm{eff}}(\mathbf{k})) \tag{3}$$

with

$$H_{\mathrm{eff}}(\mathbf{k}) = i/T \ln_{-\psi} U(t = T, \mathbf{k}), \tag{4}$$

where $-\psi$ denotes the branch-cut of the logarithm. The procedure to define such an operator $V_\psi$ and thus the invariant $W_\psi$ for discrete-time evolutions (that is, when the dynamics is given by a succession of scattering events and where time therefore does not appear explicitly), as in our model, was developed in a previous detailed study[33] (in particular in sections V.A. and V.B.).

Importantly, the branch-cut $\psi$ must be chosen in a spectral gap of $U(T, \mathbf{k})$, or $S(\mathbf{k})$ in our case. For this reason, $W_\psi$ is said to be a gap invariant, and indeed directly gives the number of chiral edge states in a given quasi-energy gap $\psi$. In contrast, Chern numbers are band invariants. They are inferred from the eigenstates of $H_{\mathrm{eff}}(\mathbf{k})$ expressed in equation (4) and thus cannot capture the full unitary evolution. Finally, the details for the calculation of the invariants $W_\psi$ in oriented kagome graphs can be found in Delplace et al.[33]. Their values for the band structures of Fig. 1c are 1,1,1,1,1,1 in the anomalous case and 1,0,1,1,0,1 for the Chern case. For completeness, we provide the bandgap map of the network together with the values of the homotopy invariant in Supplementary Fig. 8.

### Simulations

The simulation method of arbitrary finite non-reciprocal honeycomb networks is based on the scattering matrix method. For a finite non-reciprocal network with $N_r$ input/output ports, once we have the information of the scattering matrix of each non-reciprocal element and the distribution of the phase delays of the links, this method can provide (i) the scattering matrix $S_{N_r}$ regarding the $N_r$ port system, and (ii) the field map across the network knowing the excitations at the $N_r$ ports (see details in Supplementary Information part II).

We exemplify this method by calculating the transmission between 'Geneva' and 'Davos' through the Switzerland-shaped network (the

network used in Fig. 4 of the main text) as a function of $\varphi$, and compare the transmission results with the ribbon band structures (see Supplementary Fig. 2). We assume a uniform distribution for the phase delay $\varphi$ and the same non-reciprocal elements (in anomalous or Chern phase) in the Switzerland-shaped network. When both anomalous and Chern phases fall in a topological bandgap, the transmission is near unity. When both phases fall in a bulk band, the transmission undergoes sharp variations with $\varphi$, depending on the excited bulk mode. Only the Chern phase exhibits bands of blocked transmission, owing to the trivial bandgaps.

### Design

The non-reciprocal networks are designed and fabricated on 0.508 mm thick Rogers RT/duroid 5880 substrate (dielectric loss $\tan\delta = 0.0009$ at 10 GHz) with 35 μm thick copper on each side. Here, the non-reciprocal element is a surface mount microwave circulator (UIYSC9B55T6, UIY Co.), designed from a 'Y'-shaped strip line on a printed circuit board[48]. The three ports are placed 120° apart from each other such that they are iso-spaced. The printed circuit board is sandwiched between two pieces of ferrite. Without magnetic fields, the 'Y'-junction strip line supports two degenerate modes at $\omega_0$: right-handed and left-handed. To bias it, two magnets are fixed outside, providing the required magnetic field of $50\,\mathrm{kA\,m^{-1}} = 628$ Oe, normal to the printed circuit board and polarizing the ferrite, therefore lifting the initial degeneracy, with chiral modes at $\omega_+$ and $\omega_-$. In our experiment, we first measure an individual circulator and retrieve its scattering matrix $S_0$. The measured reflection of an individual circulator is shown in Extended Data Fig. 4a, and sets the frequency bands for CI and AFI operations.

Microstrip lines serve as phase delay links, with a width of 1.65 mm, corresponding to a standard 50 ohm characteristic impedance. The phase delay $\varphi$ induced by a microstrip line with length $L$ operating at frequency $f$ is expressed as $\varphi = (2\pi L f \varepsilon_{\mathrm{eff}}^{1/2})/c$, where $\varepsilon_{\mathrm{eff}}$ is the effective permittivity of the microstrip line, and can be determined by an empirical formula[49]. Taking into account the frequency dispersion of the lines and circulators, we construct a more practical topological bandgap map, shown in Extended Data Fig. 4b, as a function of the effective length of the microstrip lines $L$ and the operating frequency $f$. With the aid of the map, we select $L_1 = 26.5$ mm and $L_2 = 37.5$ mm, which produce the conditions $\varphi = \pi/8$ and $\varphi = \pi/2$, respectively, in the simulations (Fig. 3a, Extended Data Figs. 5a, 6a). As exhibited in Extended Data Fig. 4c, the fabricated networks show the microstrip lines of $L_1$ (blue dashed region) and $L_2$ (red dashed region).

### Measurements

The scattering parameters and field maps of three fabricated networks (network 1, network 2 and the Switzerland-shaped network) are measured by a vector network analyser (VNA; ZNB20, R&S), as demonstrated in Extended Data Fig. 8. For the scattering parameter measurements (Extended Data Fig. 4), as the networks are multiport, we connect the two ports of the VNA to two ports of the measured network, with the other network ports perfectly matched with 50-ohm terminations (no reflection). For the longer-range transport measurement shown in Extended Data Fig. 7, we connect ports 1 and 4 to the two VNA ports, while letting ports 2 and 3 be open (full reflection) and perfectly matching ports 5 and 6. For the field map measurements, we connect the signal input port of the measured network to VNA port 1, while perfectly matching the other ports of the network. We manually probe the field at the middle of the microstrip lines by using a coaxial probe, which is connected to VNA port 2, as shown in Extended Data Fig. 8b.

### Validation of the model assumptions

The model is the one of a unitary scattering network, namely, lossless scatterers connected by links imparting phase delays. Microstrip transmission lines are known to behave as pure phase delays in this

frequency range, since the propagation losses over so short distances are negligible (we indeed measured them to be 0.0167 dB cm$^{-1}$). We are therefore left with checking that Supplementary equations (1)–(3) (see details in Supplementary Information part II) are a good model for the scatterers.

We start by checking the validity of the assumptions behind Supplementary equations (1)–(3), namely, that the scatterers have three-fold rotational symmetry (C3 symmetry), and that they are unitary. To do this, we measured the scattering matrix $S^M$ of our scatterers. We start with checking C3 symmetry, which implies that $S_{12} = S_{23} = S_{31}$, as well as $S_{11} = S_{22} = S_{33}$. Extended Data Fig. 3a plots the moduli and arguments of all these quantities in the considered frequency range. From these plots, we see that although some small deviations from C3 symmetry are observed in the reflection coefficients, they correspond to fluctuations of reflection below −20 dB. We conclude that C3 symmetry is a valid assumption.

Next, we check unitarity. Extended Data Fig. 3b plots the eigenvalues of the measured scattering matrix versus frequency, in the complex plane. We can see that they are always very close to the unit circle, meaning that unitarity is also a very reasonable assumption. This is expected since we used a substrate with a small loss tangent of $10^{-4}$ and circulators with low insertion losses of 0.2 dB. Absorption is therefore not expected to alter the prediction of the unitary theory, but simply to add an exponential decay which shows itself especially for large samples. For example, while long range transport from Geneva to Davos in the circulator network of Fig. 4b is associated with 20 dB of signal attenuation, the presence of the edge mode predicted by the unitary theory is not affected (see Extended Data Fig. 7).

Now, we estimate the error that we make by modelling the real matrix $S^M$ with Supplementary equations (1)–(3). To do this, we find the C3-symmetric unitary scattering matrix $S^U$ that is the closest to $S^M$. We get $S^U$ by rescaling the eigenvalues of $S^M$ to make them exactly unitary, keeping their arguments. We then determine the parameters $\xi$ and $\eta$ of $S^U$, which we plot against frequency in Extended Data Fig. 3c. We then define an S-parameter error metric as

$$\varepsilon = \left[ \frac{1}{3} \left( |S_{11}^M - S_{11}^U|^2 + |S_{12}^M - S_{12}^U|^2 + |S_{21}^M - S_{21}^U|^2 \right) \right]^{1/2}. \tag{5}$$

This quantity represents the error that we make by using Supplementary equations (1)–(3). It is plotted in Extended Data Fig. 3d. We see that this error is below 5% at all frequencies, which unambiguously validates the relevance of Supplementary equation (3).

## Data availability

The data that support the findings of this study are available at https://doi.org/10.5281/zenodo.5101825.

## Code availability

The codes that support the findings of this study are available at https://doi.org/10.5281/zenodo.5101825.

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

**Acknowledgements** Z.Z. and R.F. acknowledge funding from the Swiss National Science Foundation under the Eccellenza award number 181232. P.D. acknowledges support from the IDEX Lyon Breakthrough programme ToRe (contract no.ANR-16-IDEX-0005).

**Author contributions** Z.Z. performed the numerical simulations and experiments, under the supervision of R.F. P.D. and R.F. conceived the project. All authors participated in writing and revising the manuscript.

**Competing interests** The authors declare no competing interests.

**Additional information**
**Correspondence and requests for materials** should be addressed to Romain Fleury.

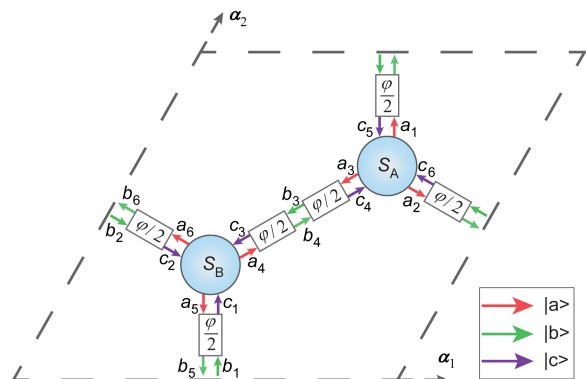

**Extended Data Fig. 1 | Detailed schematic of the unit cell of the non-reciprocal network and signal labelling convention.** We define three state vectors: $|a>$, $|b>$, and $|c>$, which represent scattering wave amplitudes propagating out, between and into the non-reciprocal elements, respectively. The total phase delay between two scatterers is $\varphi$.

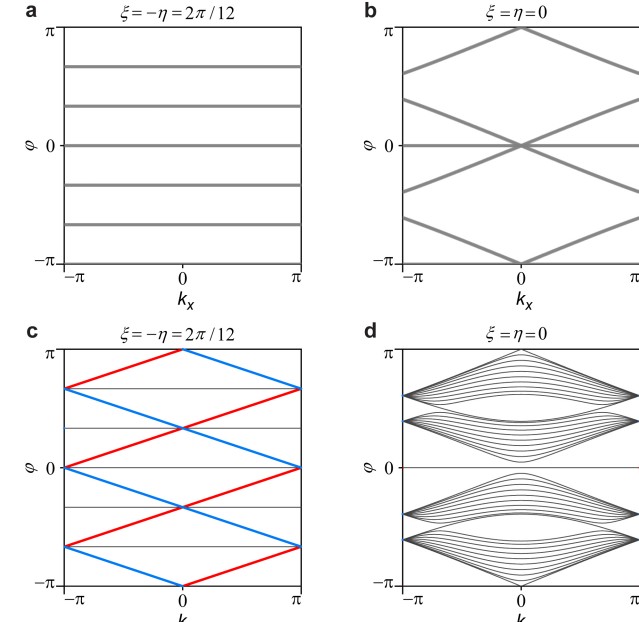

**Extended Data Fig. 2 | Floquet band structures at two special points of the topological phase diagram. a, b**, Bulk band structures at the green (**a**) and centre (**b**) points of the phase diagram of Fig. 2c in the main text. The green point corresponds to a phase-rotation symmetric network of perfect matched circulators, thus in AFI phase. The red centre point represents a network of reciprocal resonant scatterers, with all bandgaps closed. **c**, **d**, Ribbon band structures corresponding to panel **a** and **b**, respectively. The perfect circulator network features flat bulk band with dispersionless edge modes regardless of the value of the quasi-energy $\varphi$, which can only occur in the AFI phase.

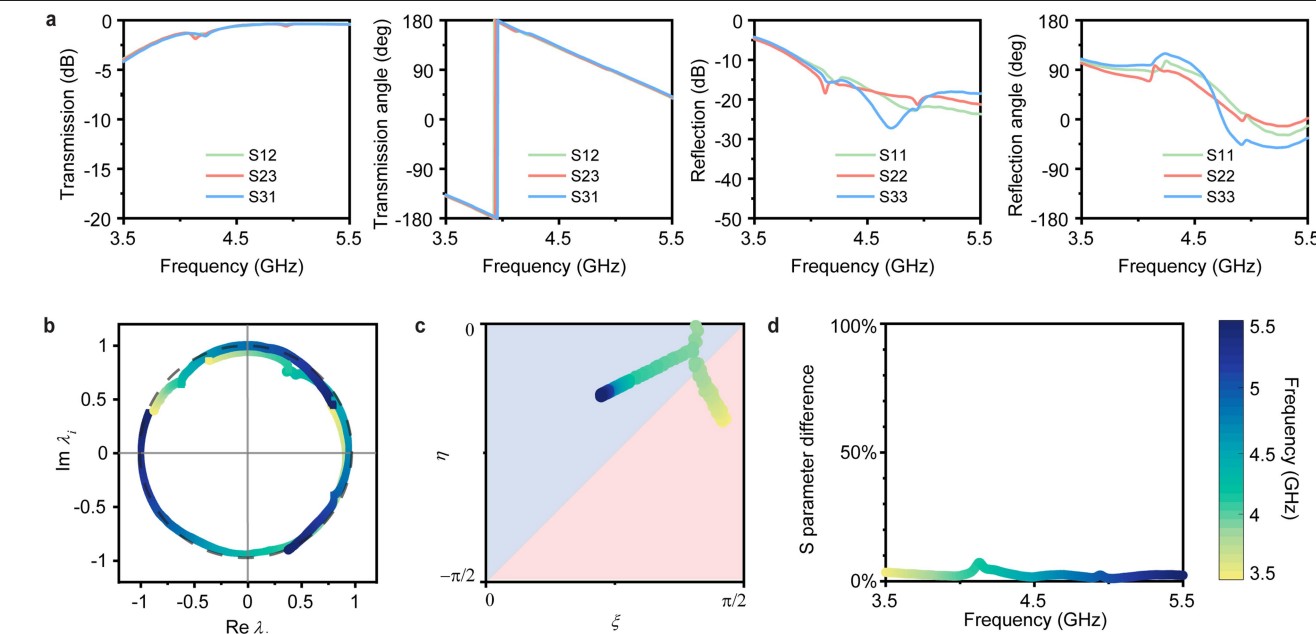

**Extended Data Fig. 3 | Experimental validation of the model assumptions.**
**a**, C3 symmetry holds when $S_{12} = S_{23} = S_{31}$ as well as $S_{11} = S_{22} = S_{33}$, which is very well satisfied in the considered frequency range. **b**, Eigenvalues of the measured scattering matrix, with nearly-unitary behaviour over the entire experimental bandwidth. **c**, $\xi$ and $\eta$ parameters used to approximate the real scattering matrix with a C3-symmetric unitary matrix. The red area is the Chern phase, and the blue the anomalous one. **d**, Error in % made by approximating the real scattering matrix with equation (4) over the entire bandwidth.

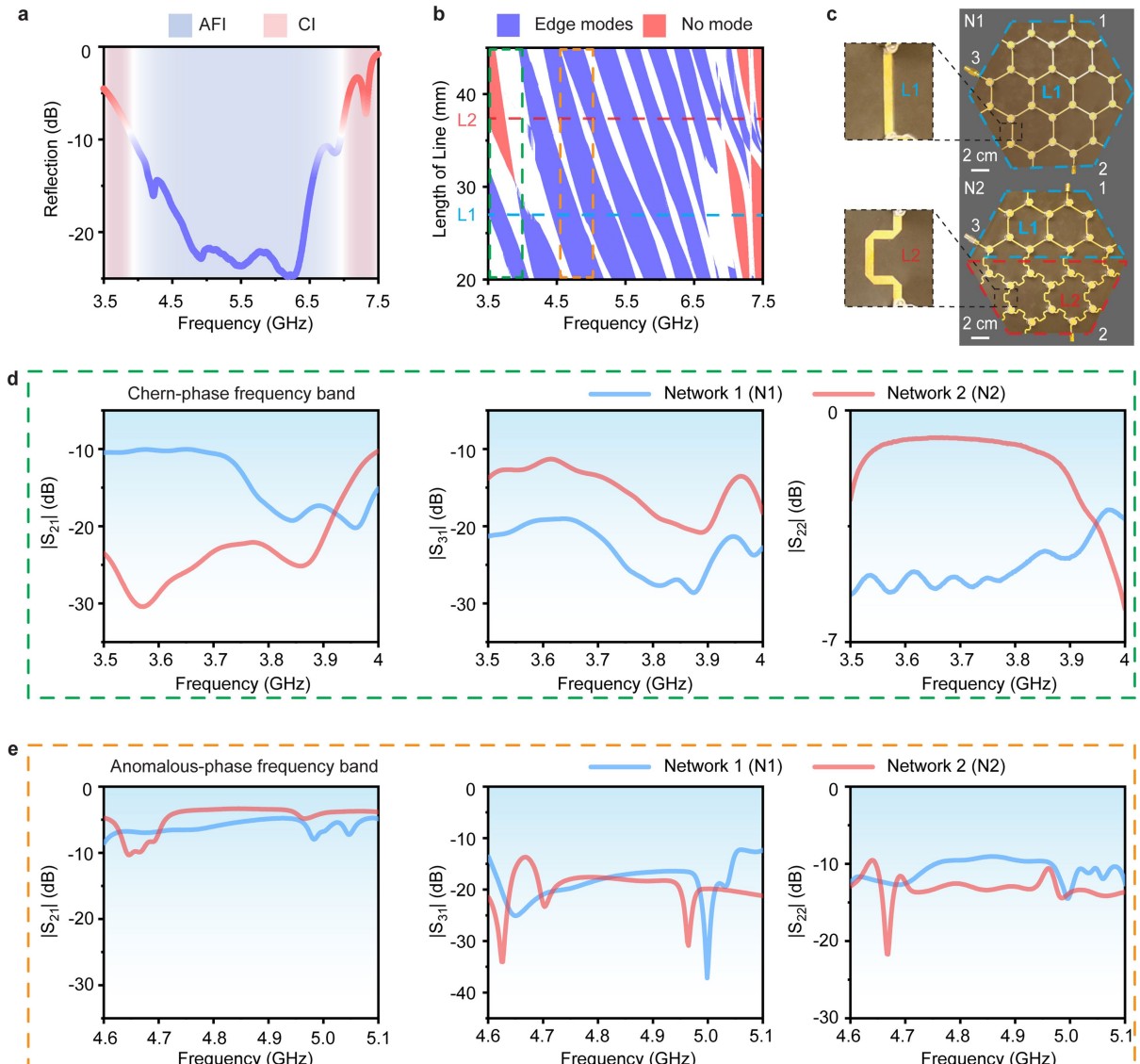

**Extended Data Fig. 4 | Experimental network design and measured scattering parameters. a**, Measured reflection spectrum of an individual ferrite circulator. The blue-shaded area represents the bandwidth of the anomalous phase, corresponding to low reflection ($|R| < -9.5$ dB $= 20 \cdot \log_{10}(1/3)$). By contrast, the red-shaded area shows the Chern phase with high reflection ($|R| > -9.5$ dB). Topological phase transitions happen at around 3.9 GHz and 7 GHz. **b**, Topological bandgap map predicted from the individual scattering data, when varying the length of the microstrip connections and the operating frequency. The blue and red regions correspond to bandgaps with and without topological edge modes, respectively. The white regions represent bulk bands. **c**, Design details of the experimental networks probed in Fig. 3b of the main text. Network 1 (N1) has a uniform length distribution of microstrip lines with $L = L1$. For network 2 (N2), we introduce an interface and replace the bottom part with lines of different length $L2$. **d**, Measured amplitudes of the scattering parameters $S_{21}$ (left), $S_{31}$ (middle) and $S_{22}$ (right) in the Chern-phase frequency band (green dashed box in panel **b**). **e**, Measured scattering parameters in the anomalous-phase frequency band (yellow dashed box in panel **b**).

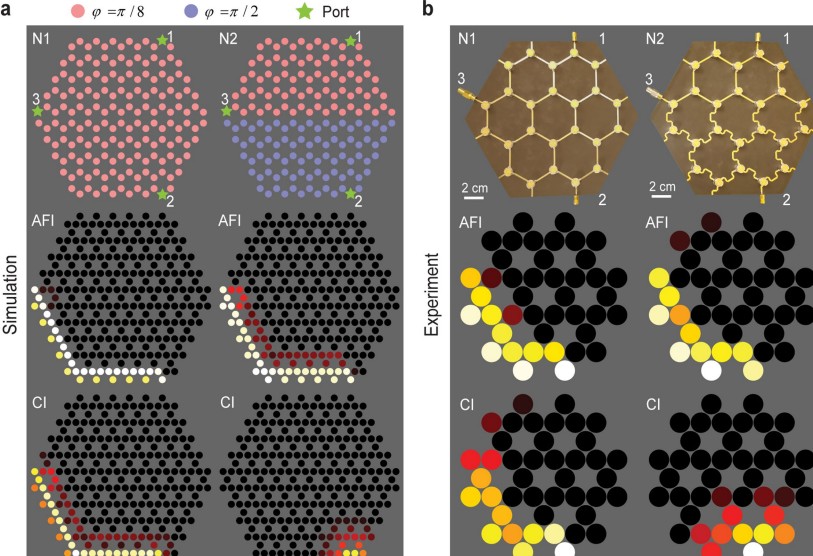

**Extended Data Fig. 5 | Numerical and experimental field maps for excitation at port 2. a**, Numerical predictions for excitation at port 2 for the same system as in Fig. 3 of the main text. While the anomalous phase supports transmission to port 3 regardless of the phase link distribution, the Chern phase possesses a trivial bandgap at $\varphi = \pi/2$, and reflects all the energy incident from port 2, see bottom right plot (the field distribution exhibits exponential decay). **b**, Corresponding experimental data.

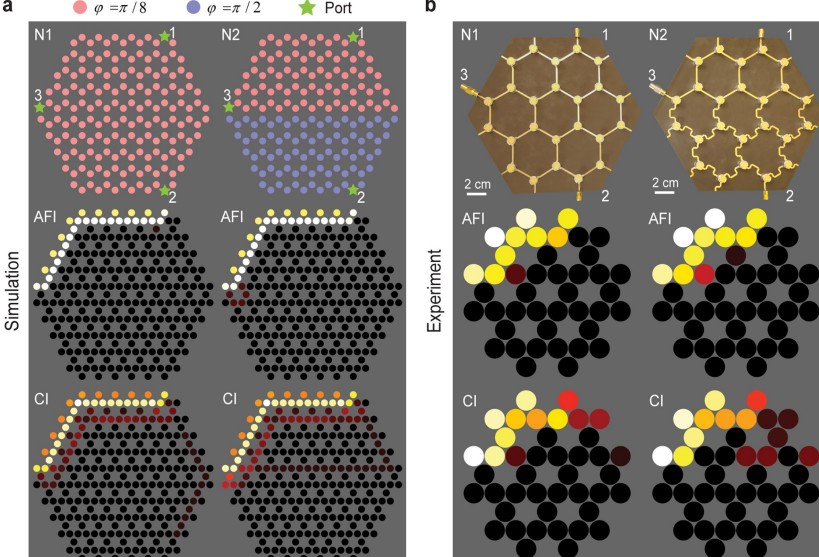

**Extended Data Fig. 6 | Numerical and experimental field maps for excitation at port 3. a**, Numerical predictions for excitation at port 3 for the same system as in Fig. 3 of the main text. Both the anomalous and Chern phases fall in topological bandgap at $\varphi = \pi/2$, leading to transmission to port 1. **b**, Corresponding experimental data.

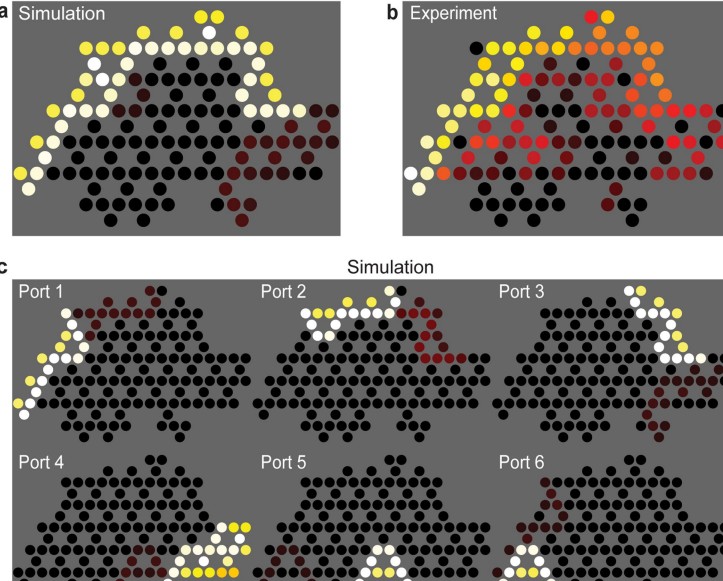

**Extended Data Fig. 7 | Additional field maps for the anomalous topological Switzerland-shaped network.** We plot simulated (**a**) and experimental (**b**) transmissions from Geneva (port 1) to Davos (port 4) for the same network in Fig. 4 of the main text, leaving all other ports open. **c**, Numerical prediction corresponding to the experimental data shown in Fig. 4c of the main text.

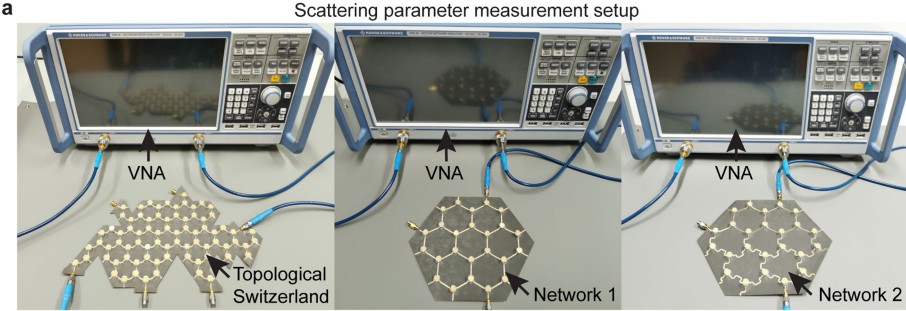

**a** Scattering parameter measurement setup

VNA VNA VNA

Topological
Switzerland Network 1 Network 2

**b** Field map measurement setup

Probe

**Extended Data Fig. 8 | Experimental setups for scattering parameter and field distribution measurements. a**, The setup consists of a vector network analyser (VNA) and three microwave non-reciprocal networks: the Switzerland-shaped network (left), N1 (middle), and N2 (right). **b**, Field map measurement with a coaxial probe for measuring fields on the microstrip lines.

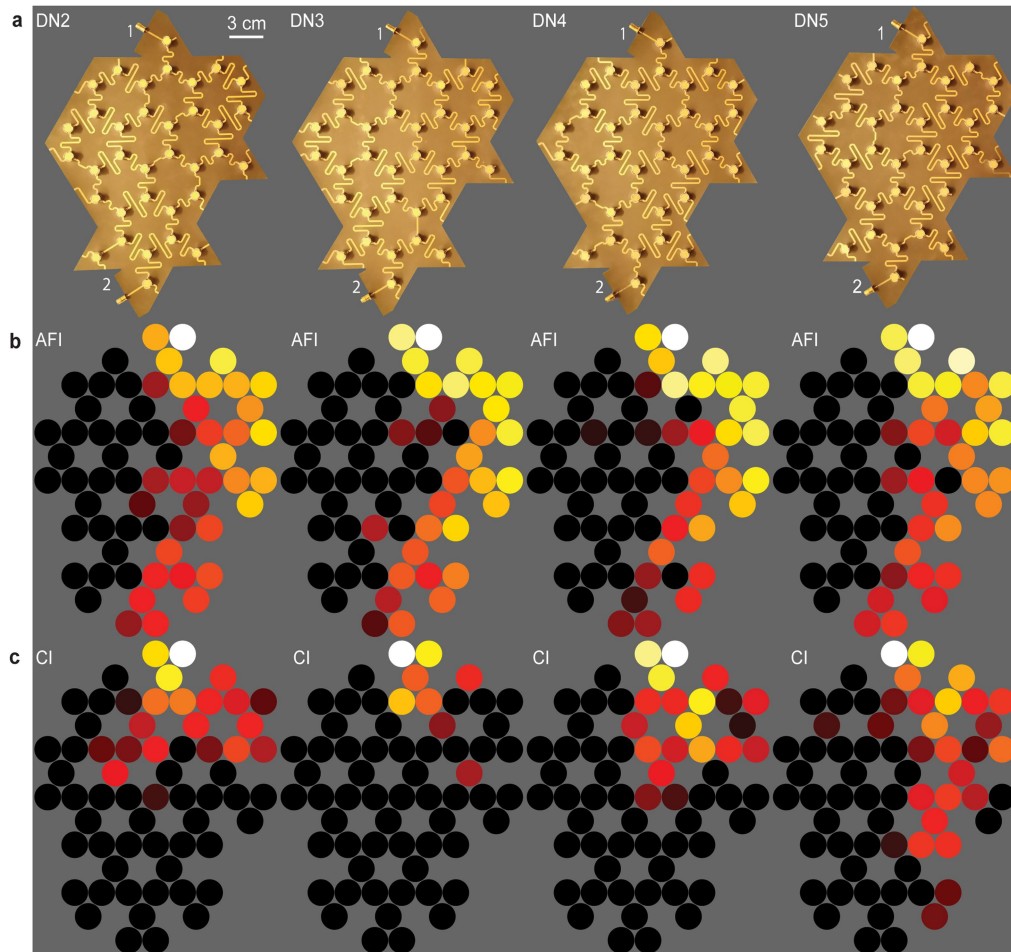

**Extended Data Fig. 9 | Experimental validation of anomalous phase disorder robustness in four other prototypes with distinct disorder realizations. a**, Pictures of the prototypes, having the same irregular shape but different phase delay distributions implemented by varying the geometry of the serpentine links. **b**, Measured field maps in the AFI phase. **c**, Measured field maps in the CI phase.