## [Peer Review File · Nature]

Manuscript Title:

Superior robustness of anomalous nonreciprocal topological edge states

Editorial Notes:

Reviewer Comments & Author Rebuttals

Reviewer Reports on the Initial Version:

Ref #1

In "Anomalous nonreciprocal topological networks: stronger than Chern insulators", the authors experimentally realized a microwave circulator network that supports anomalous Floquet Chern modes, and showed that the robustness of its topological transport is extremely high, at least for disorder in phase delays. Their experiment is well-motivated by their theoretical simulations, where they set up a network, employ a scattering matrix formalism and converted the model into a Floquet periodic model.

In terms of novelty, the experimental demonstration indeed seems to be the best realization of anomalous Floquet topological modes with microwave networks. Conceptually, however, it does not showcase new physics, being based on well-known mappings (ie Ref 32) of such setups to a Floquet Bloch system and well-known concepts of anomalous topological modes (ie Ref 42).

After reading this manuscript, many aspects remain unclear. I will be in better position to make a recommendation only after the authors respond to the following queries/comments:

1. The presentation of the model is very unclear. In the main text, mention was made of parameters like zeta and eta, etc, but no physical insight about them were given. Indeed, the definitions of model and setup are distributed across the main text, main text images, methods, and extended data etc, and that makes the paper tiring to read. Even after careful reading, some aspects are still unclear. For instance, Fig 1c looks quite schematic. Is it based on actual computations? How do we trust that Eq 2 should describe the system accurately in principle, other looking at the comparison between experiment and theory in Fig 3? I know that the theory was already given in Ref 32, but more precise justification should be given when one applies the theory to a real setup.

2. The authors should explain why their experiment is a breakthrough compared to other experiments realizing similar anomalous Floquet modes (and thus exhibiting the same type of robustness in principle). In particular, why is it an engineering challenge? It seems that the lower the reflection coefficient R , the more robust the anomalous Floquet phase, so why not just set $R=0$? Or is low R the true challenge?

3. Since this is already a network of circulators, is it really that surprising that it allows robust chiral propagation? As someone who also works on topological systems, I may be inclined to welcome experimental demonstrations of topology as novel. However, this may not come naturally to other Nature readers, and the authors should explain that more.

4. The authors showed that their network is able to withstand disorder of much larger magnitudes than the bandgap. But their demonstrated disorder seems to be restricted to phase disorders. In other words, "much larger than the band gap" is not really a statement of comparison about magnitudes. I invite the authors to answer: what types of phase disorder can destroy their anomalous Floquet topological modes?

5. Related to the point above, it will be important to also consider disorder in magnitudes of the couplings, and not just their phases i.e. circulator geometry, gain/loss etc. Does the network then map to a non-reciprocal non-Hermitian system? If yes, such systems will also experience the non-Hermitian skin effect, and that will be an even more robust phenomenon that has to be treated. In fact, the non-Hermitian skin effect is at least as inevitable in the presence of non-hermiticity and non-reciprocity as robust chiral propagation is in the presence of non-reciprocity.

Ref #2

The manuscript by Zhang et al deals with a type of anomalous nonreciprocal networks which has stronger robustness against defects than the Chern insulators. The authors establish the model of nonreciprocal topological networks, and set the reflection or the degree of nonreciprocity for one unit-cell as an indicator of different topological phases of the network. Further, the authors find that when the reflection is below a threshold or the degree of nonreciprocity for the building block is large enough, the system will support a type of anomalous edge mode that has superior robustness over the Chern one against the phase link disorder of an arbitrarily large value. The authors proved by simulations and experiments that the proposed structure can achieve good one-way wave transport.

After going through this work, I have the following concerns which requires the authors to clarify

1. The first concern is about the definition of anomalous phase, phase transition and Chern phase. The similar nonreciprocal topological networks have been constructed previously, for example, in acoustics (Phys. Rev. Lett. 122, 014302, 2019, not cited in this work). The main discovery here is that the degree of nonreciprocity or the reflection of one unit-cell can be tailored to drive the network into two different topological phases. For example, the low reflection or high degree of nonreciprocity case is termed as an anomalous phase, with the quasi-energy band structure featured with the topological gapless edge modes in all band gaps. The high reflection or low degree of nonreciprocity case is termed as the Chern phase, with the quasi-energy band structure featured with some band gaps trivial of no topological edge modes. However, in PRL 122, 014302, 2019, the discussed case seems like to be the anomalous phase in this work, where the paper named it as the Chern phase and the network as acoustic Chern insulator, since the Chern number is calculated to be non-zero. So I have a question that what is the topological invariant or Chern number of the bands in Fig. 1c?

2. The superior robustness of the anomalous phase can be well understood from the quasi-energy band structures in Fig. 2. After a further thought, it can be found that the superior robustness is specific to the link-phase variations but not to the structure of the circulators or the operation frequency as shown in the Extended Data Fig. 4a. Intuitively, the link-phase variation does not change the degree of nonreciprocity for each unit-cell in the network. Also for the AFI case, the superiority is limited by the existence of bulk bands in quasi-energy band structures, even though the case of CI is worsen due to the existence of additional band gaps.

3. From the Extended Data Fig. 7b, the experimental measurement shows an obvious field attenuation in comparison to simulation. For a much larger network in practical use, the practicability test would fail if the field attenuation arises. In my viewpoint, the wave attenuations due to the reflection/scattering and the absorption are equally important to be solved.

4. I found there may be a typo in the manuscript. In the Measurement setup part, it should be "we connect ports 1 and 4 to the two VNA ports, while letting ports 2 and 3 open."

Besides those concerns, I also have some suggestions for the authors:

1. For Fig. 4, it will be more interesting if the authors can introduce some link-phase disorders into the Switzerland shaped network.

2. In Fig. 2b, the authors can show the top edge mode and bottom edge mode for the three cases in Fig. 2a. I am wondering whether there exists any difference in field distributions among the edge modes for the anomalous phase, phase transition point and Chern phase.

3. For Extended Data Fig. 3, the authors can show the measured transmission for AFI and CI to verify the simulation result.

Overall this work strikes me as an interesting concept and a nice experimental demonstration, but both the theoretical and experimental validation in the current form seems slender to merit publication in Nature.

Ref #3

In the manuscript titled "Anomalous nonreciprocal topological networks: stronger than Chern insulators", the authors have reported on the design and realization of a time-reversal symmetry-breaking anomalous Floquet insulator (AFI). By changing the transmission coefficient, this system can be switched between the AFI phase, the Chern insulator (CI) phase, and the normal insulator phase. The authors confirm that AFI is more resistant to the phase link disorder compared to the Chern insulator phase.

I think that the system proposed by the authors, which is insensitive to disorder and defects, is an interesting physical system and has potential applications. However, the current paper lacks a theoretically rigorous derivation to demonstrate that the AFI is more robust to impurities and defects than the CI. In the current state, I do not recommend the article to be published in Nature.

I would like the authors to consider the following questions related to their experiment and analysis.

1) The authors compared the AFI state with the CI state and chose the CI state which has boundary states in band gap type 1 but not in band gap type 2. While the AFI state has boundary states in both types of gaps. This is one of the main reasons why the transport properties of the AFI boundary state can resist stronger phase link disorder than CI. However, if the appropriate parameters are chosen or the system is well-designed, it is possible to make the boundary states exist in all gaps of the CI state. For this type of CI state, it should have the same robustness to phase link disorder as AFI. Therefore, the authors' evidence that AFI is more robust to phase disorder than CI is not very convincing and rigorous. The authors should better prove rigorously that AFI has a certain universal physical mechanism which makes it more robust to phase link disorder than CI.

2) The authors report that the boundary transport properties of the AFI system are robust to the disorder of the phase ϕ . Is this related to the fact that ϕ appears as an exponent in $\exp(-\phi)$ in the eigenvalue equation? Does this form of eigenvalues result in more stability for impurities and defects than the ordinary form of eigenvalue equation? For other types of impurities, such as lattice distortion or the disorders on ω_{pm} , does the system possess a similar high degree of robustness.

3) Which class does the AFI belong to according to the classification of topological insulators? Which topological number is used to characterize its topological properties? How does the topological number change when changing from AFI state to CI state? Can a more detailed form of $S(k)$ be given in Equation 1? So that it can be clearly distinguished whether the system is in the CI or AFI state. Can the authors give the topological energy gap as a function of phase links and the formula for the topological number?

4) How strong does the external magnetic field required to break the time-reversal symmetry?

Author Rebuttals to Initial Comments:

Reviewer 1

R1.1: In "Anomalous nonreciprocal topological networks: stronger than Chern insulators", the authors experimentally realized a microwave circulator network that supports anomalous Floquet Chern modes, and showed that the robustness of its topological transport is extremely high, at least for disorder in phase delays. Their experiment is well-motivated by their theoretical simulations, where they set up a network, employ a scattering matrix formalism and converted the model into a Floquet periodic model. In terms of novelty, the experimental demonstration indeed seems to be the best realization of anomalous Floquet topological modes with microwave networks.

Response: We thank the Reviewer for his/her careful review and positive comment about our experiment.

R1.2: Conceptually, however, it does not showcase new physics, being based on well-known mappings (i.e. Ref 32) of such setups to a Floquet Bloch system and well-known concepts of anomalous topological modes (i.e. Ref 42).

Response: Beyond the originality of our experiment, which reports a genuinely non-reciprocal anomalous Floquet insulator, we report a key conceptual discovery: the extraordinary robustness of edge mode transmission in the anomalous phase. Topological wave transmission that can survive distributed disorder of strength larger than the band gap size is a remarkable physical property with strong potential applications that has never been discussed or demonstrated before.

Here, we report and demonstrate experimentally this unique feature of nonreciprocal anomalous networks. We stress that Refs 32 and 42, which are two theoretical works, do not talk at all about disorder or the transmission properties of the anomalous and Chern phases.

Revision: The conceptual novelty of our findings is stressed in the introduction, and in the discussion of the corresponding key results, related to Fig. 3.

R1.3: After reading this manuscript, many aspects remain unclear. I will be in better position to make a recommendation only after the authors respond to the following queries/comments:

Response: We are happy to clarify the points raised by the Reviewer. We thank the Reviewer for all the relevant remarks that have led to additional studies and helped us improve the presentation of our findings.

R1.1.4: The presentation of the model is very unclear. In the main text, mention was made of parameters like ζ and η , etc, but no physical insight about them were given. Indeed, the definitions of model and setup are distributed across the main text, main text images, methods, and extended data etc, and that makes the paper tiring to read.

Response: We apologize if the initial presentation of our model was not optimal, and we are happy to take this opportunity to revamp it. Given the length limitations of the journal, we cannot provide all the details about the modeling in the main text, however we committed to make it easier to read, while conveying our main message in the best possible way.

With the comment of the Reviewer, we realized that the initial version of our main text did not provide enough physical insight about the parameters ξ and η , and their connection to physical quantities like reflection level or non-reciprocal isolation. Because of this, the Reviewer was unnecessarily pushed to the Methods and Supplementary, where the information was not easy to find.

The important point is that using the angles ξ and η is more rigorous than translating the parameters to more intelligible quantities, like reflection and non-reciprocal isolation, because these parameters contain all information about 3-by-3 unitary matrices with C3 symmetry, including all possible combinations of the scattering phases. That being said, we are happy to provide information about how ξ and η map into the reflection and non-reciprocity level of the circulators in the main text.

Revision: We have avoided unnecessary use of the angular parameters in the main text. In addition, we have revised Fig. 2c to include an extra panel showing the correspondence between ξ and η and the level of non-reciprocity of the non-reciprocal scatterers (the isolation $|S_{21}/S_{12}|$). The two panels of Fig. 2c therefore show the full mapping between ξ and η and important physical parameters, namely the reflection and non-reciprocal isolation.

R1.5: Even after careful reading, some aspects are still unclear. For instance, Fig 1c looks quite schematic. Is it based on actual computations?

Response: Yes, Fig. 1c is actual data coming from computations. Notice that the bulk band structures of Fig. 1c and the ribbon band structures of Fig. 2a are consistent, as expected.

Revision: We have revised the main text so that it is very clear that the semi-analytical model is used to generate the data in Fig. 1c and Fig. 2a.

R1.6: How do we trust that Eq 2 should describe the system accurately in principle, other than looking at the comparison between experiment and theory in Fig 3? I know that the theory was already given in Ref 32, but more precise justification should be given when one applies the theory to a real setup.

Response: We are happy to provide more direct evidence that the model applies to the real setup.

The model used in the paper is the one of a unitary scattering network, namely lossless scatterers connected by links imparting phase delays. There is no question about the validity of modeling microstrip transmission lines as pure phase delays in this frequency range, since the propagation losses over so short distances are negligible (we indeed measured them to be 0.0167 dB/cm). As the Referee points out, we are therefore left with checking that Eqs. 2-4 are a good model for our scatterers.

We start by checking the validity of the assumptions behind Eqs. 2-4, namely that the scatterers have three-fold rotational symmetry (C3 symmetry), and that they are unitary. To do this, we measured the scattering matrix S^M of our scatterers. We start with C3 symmetry, which implies that $S_{12} = S_{23} = S_{31}$, as well as $S_{11} = S_{22} = S_{33}$. Fig. R1a plots the moduli and arguments of all these quantities in the considered frequency range. From these plots, we see that although some small deviations from C3 symmetry are observed in the reflection coefficients, they correspond to fluctuations of reflection below -20dB. We conclude that C3 symmetry is a valid assumption.

Next, we checked unitarity. Fig. R1b plots the eigenvalues of the measured scattering matrix versus frequency, in the complex plane. We can see that they are always very close to the unit circle, meaning that unitarity is also a very reasonable assumption.

Now, we directly address the comment of the Reviewer. We estimate the error that we make by modeling the real matrix S^M with Eqs. 2-4. To do this, we find the C3-symmetric unitary scattering matrix S^U that is the closest to S^M . We get S^U by rescaling the eigenvalues of S^M to make them exactly unitary, keeping their arguments. We then determine the parameters ξ and η of S^U , which we plot against frequency in Fig. R1c. We then define an S-parameter error metric as

$$\varepsilon = \sqrt{\frac{|S_{11}^M - S_{11}^U|^2 + |S_{12}^M - S_{12}^U|^2 + |S_{21}^M - S_{21}^U|^2}{3}}$$

This quantity that represents the error we make by using Eqs. 2-4, is plotted in Fig. R1d. We see that this error is below 5% at all frequencies, which unambiguously validates the relevance of Eq. 2.

Figure R1: Evidence that the measured scattering matrix can be approximated as C_3 -symmetric and unitary, and that our model can be applied. **a**, C_3 symmetry holds when $S_{12} = S_{23} = S_{31}$, as well as $S_{11} = S_{22} = S_{33}$, which is very well satisfied in the considered frequency range. **b**, Eigenvalues of the measured scattering matrix, with nearly-unitary behavior over the entire experimental bandwidth. **c**, ξ and η parameters used to approximate the real scattering matrix with a C_3 -symmetric unitary matrix. The red area is the Chern phase, and the blue area the anomalous one. **d**, Error in % made by approximating the real scattering matrix with Eq. 2 over the entire bandwidth.

Revision: We have added this justification to the Methods section and Fig. R1 above as a new Extended data figure 3.

R1.7: 2. The authors should explain why their experiment is a breakthrough compared to other experiments realizing similar anomalous Floquet modes (and thus exhibiting the same type of robustness in principle).

Response: We believe that our experiment and results are unmatched because:

- 1) There exist no experiments of electromagnetic Floquet phases involving genuinely non-reciprocal transport.
- 2) Prior experiments have not considered the effect of *distributed* random disorder on the edge mode transport. Conceptually, the superiority of the anomalous phase has never been established before.
- 3) Even for Chern wave insulators, prior experimental investigation of backscattering immunity has been limited to single, isolated defects.
- 4) Our experiment has large applicative potential in microwaves, because our platform is fully compatible with standard transmission lines (microstrip technologies), off-the-shelf surface mount components (circulators), and standard fabrication techniques (PCB manufacturing).

Revision: Point 1) is stressed below Eq. 1. Points 2) to 4) are now stressed in the conclusion.

R1.8: In particular, why is it an engineering challenge?

Response: We do not claim that our experiment is a particular engineering challenge. Instead, it is a simple tabletop experiment that can be easily reproduced. We believe that this is an undeniable advantage of our approach, and not a disadvantage. Indeed, the compatibility with standard transmission lines, surface mount components and scalable manufacturing techniques makes our platform ideal for real-life applications in microwave technologies and communication systems.

Revision: This point has been added to the conclusion.

R1.9: It seems that the lower the reflection coefficient R , the more robust the anomalous Floquet phase, so why not just set $R=0$? Or is low R the true challenge?

Response: It is correct that $R=0$, which corresponds to a phase-rotation symmetric point², is the best design. It implies perfect matching of the circulators. Although we could closely approach it, we do not need to enforce such a critical condition to leverage the exceptional robustness of the anomalous phase. Indeed, Fig. 3c (left) in the main text is generated for 16% of reflection, and the transmission remains above 90% regardless of the strength of the phase delay disorder. The unnecessary to set the reflection to a particular, critical value, is a very practical advantage of the method.

Revision: We comment on this point in the discussion of Fig. 3 in the revised version of the manuscript.

R1.10: 3. Since this is already a network of circulators, is it really that surprising that it allows robust chiral propagation? As someone who also works on topological systems, I may be inclined to welcome experimental demonstrations of topology as novel. However, this may not come naturally to other Nature readers, and the authors should explain that more.

Response: What is surprising is not that the network supports chiral propagation, but that the chiral edge modes comes in two topologically distinct classes – Chern and anomalous – which do not contribute in the same way to edge transport when random distributed disorder is added. The fact that it is possible for anomalous topological edge transport to survive disorder of magnitude larger than the quasi-energy band gap is truly remarkable and has never been evidenced before. Thus far, Chern insulators were considered as providing the strongest possible level of topological protection for wave transport: we show here that it is not the case.

Revision: We have edited the introduction and conclusion of the paper so that the novelty clearly stands out.

R1.11: 4. The authors showed that their network is able to withstand disorder of much larger magnitudes than the bandgap. But their demonstrated disorder seems to be restricted to phase disorders. In other words, "much larger than the band gap" is not really a statement of comparison about magnitudes. I invite the authors to answer: what types of phase disorder can destroy their anomalous Floquet topological modes?

Response: Our statistical studies show that there exists no type of phase link disorder that can destroy the anomalous topological edge mode transmission. This is a truly unique property in anomalous non-reciprocal networks.

Indeed, for all standard topological phases, edge mode transmission is typically impeded by disorder of sufficiently large magnitude. Even in the strongest cases known, i.e. Chern insulators, on-site energy disorder of sufficiently large magnitude blocks edge mode transmission. The required disorder strength typically scales with the band gap size.

Our key point is that in anomalous non-reciprocal networks, there exist a physically relevant type of disorder that can be arbitrarily large, much larger than the band gap, without destroying edge transmission. This disorder type is phase link disorder. This is a truly remarkable and unique property, that does not exist in the Chern phase, as we prove in the paper. In the next comment, we address the only other possible disorder case for our network: disorder on the scattering matrix parameters. We also demonstrate a strikingly superior robustness of the anomalous phase over the Chern one in this case.

R1.12: 5. Related to the point above, it will be important to also consider disorder in magnitudes of the couplings, and not just their phases i.e. circulator geometry...

Response: This is a very interesting comment. Besides disorder on the phase links, one could indeed imagine another type of physically relevant disorder: on the S matrices of the scatterers themselves. Scattering matrix disorder altogether includes fluctuations of the circulator matching (reflection coefficient), non-reciprocity level, and scattering phases, which could occur due to finite geometrical tolerances in the circulator fabrication process.

In order to study this effect in a quantitative way, and compare the robustness of Chern and anomalous edge mode transmission to disorder in the scattering matrices, we performed a new statistical study. We start with the Chern phase, for which the scattering matrices must belong to the bottom right red triangle shown in the ξ and η plane in Fig. R2. We consider the same hexagonal sample as in Fig. 3a (left) of the main text, with input and output ports on the top right and bottom right corner, respectively. The ordered sample is made from uniform scatterers that are right in the middle of the Chern phase (Fig. R2a,b, left), and exhibits an edge mode along its edge, connecting the two ports (Fig. R2c, left). Turning on the disorder level to 50%, we allow the S matrices of the individual circulators to fill up 50% of the triangular area of the Chern phase. Note that this corresponds not only to reflection disorder (mismatch) but also to disorder in the degree of non-reciprocity (the isolation), as shown in Fig. R2a,b (middle column). The edge mode is already completely destroyed. The situation is even worse for fully random disorder that covers 100% of the triangular Chern phase.

Figure R2: Effect of disorder on the scattering matrix on the Chern edge mode transmission. We consider the same hexagonal network as in Fig. 3a (left) of the main text, but now each scatterer has a different scattering matrix. The left column shows the perfectly ordered system, the middle column shows a realization of random scattering disorder filling 50% of the Chern phase, and the right column shows a realization with 100% disorder, namely with scattering matrices anywhere inside the Chern phase. **a, b,** Repartition of the scattering matrices within the Chern phase (bottom right red triangle). The color map shows the corresponding reflection (panel a) and non-reciprocal isolation (panel b) values. **c,** Corresponding field maps, showing the sensitivity of Chern edge modes to scattering disorder.

Now, let us compare with the anomalous phase, for which the scattering matrices belong to the upper left blue triangle shown in the ξ and η plane in Fig. R3. By checking the excited field maps of Fig. R3c, we see a totally different behavior: the anomalous edge mode transmission is not affected much and is robust even for 100% disorder in S-parameters within the anomalous phase.

Figure R3: Same as Fig. R2 but the study is performed for the anomalous phase (top left blue triangle). Unlike the Chern one, the anomalous edge mode transmission is very robust even for fully random scattering matrix disorder.

The superiority of the anomalous phase over the Chern one in the case of scattering disorder can be quantitatively demonstrated by exploring many realizations, and performing ensemble averaging. Fig. R4 reports the result of such a statistical study, where each point of the curve is an average over 100 realizations. We see that the anomalous phase transmission can tolerate 100% disorder in the choice of scattering matrices, whereas the Chern one drops after 25% disorder level. This shows that the superior robustness of the anomalous phase is not restricted to phase link disorder, but also to the other possible source of disorder in the network: fluctuations of the scattering matrices.

Figure R4: Superior robustness of the AFI phase to scattering disorder. This new figure demonstrates that the superior robustness of the anomalous phase is not restricted to phase link disorder, but also to the other possible source of disorder in the network: fluctuations of the scattering matrices.

Revision: The main text now also includes the above statistical study demonstrating the superior resilience of the anomalous phase to scattering disorder. Fig. 3c has been revised accordingly. Two figures showing particular disorder realizations have been added to the Supplementary Information as Fig. S5 and S6.

R1.13: ... gain/loss etc.

Response: Unless we add gain on purpose (and we will consider this case in our response to the next comment R1.14), the structure we currently investigate and report can only show absorption. However, as discussed in our response above (Fig. R1b), considering the circulators as unitary scatterers is a very good approximation, because they have a very low level of insertion losses (0.2 dB). Therefore, we do not expect the real set-up to behave very differently from what the unitary theory predicts. The small losses cannot influence the presence of edge modes, only a slow decay of the edge mode amplitude is expected. This is confirmed by our experimental observations.

Revision: In the Methods, we now carefully justify why the unitary scattering theory applies to our real set-up (see the discussion related to comment R1.6). The addition of gain is discussed in the response to comment R1.14.

R1.14: Does the network then map to a non-reciprocal non-Hermitian system? If yes, such systems will also experience the non-Hermitian skin effect, and that will be an even more robust phenomenon that has to be treated. In fact, the non-Hermitian skin effect is at least as inevitable in the presence of non-hermiticity and non-reciprocity as robust chiral propagation is in the presence of non-reciprocity.

Response: The system indeed becomes non-Hermitian (the S matrices become non-unitary). The remark of the Referee is intriguing, and we are happy to provide some hints on what can happen.

We would like to discuss here a few examples, with the purpose of showing: (i) that non-Hermitian extensions of our findings do exhibit rich physics, which are too broad to be included to this paper; and (ii) that the non-Hermitian skin effect mentioned by the Referee does exist, however it does not play a critical role in edge mode transmission, and can even be completely avoided when PT symmetry is satisfied. We have to keep the discussion concise here. These effects will be the subject of future studies.

We consider the non-Hermitian perturbations represented in Fig. R5, that break our assumptions of unitarity and $C3$ symmetry. The circulators are now surrounded by non-Hermitian transmission lines that impart either local loss or gain. The circulator B is surrounded by transmission lines exhibiting exponential decay (loss) with decay factors l_1 , l_2 , and l_3 ; whereas circulator A has gain with growth factors g_1 , g_2 , and g_3 .

Figure R5: Considered non-Hermitian extension of the non-reciprocal scattering network. The circulator B is surrounded by transmission lines exhibiting exponential decay (loss) with decay factors $l_1, l_2,$ and l_3 ; whereas circulator A has gain with growth factors $g_1, g_2,$ and g_3 . Such lattice breaks the assumptions of unitarity and C3 symmetry made in our initial study.

We start by assuming no particular symmetry, taking $l_1 = l_3 = 0.2,$ $l_2 = 0,$ and $g_1 = g_2 = 0.2,$ $g_3 = 0.$ The quasi-energy band structure of a ribbon terminated at top and bottom becomes complex, and we now need to plot both real and imaginary parts of φ . We report in Fig. R6a the real and imaginary parts of the band structures when the non-Hermitian perturbations are added to a system in the anomalous phase. Fig. R6b is the case of the Chern phase. We represent in red modes that are localized to the top edge, and blue the ones localized to the bottom edge (this is detected numerically using a localization threshold). We notice that the edge modes have not been destroyed by the non-Hermitian perturbations. We also notice that the bulk modes also localize. This is the skin effect mentioned by the Referee. How do these non-Hermitian effects modify the edge transport properties? To answer this, we consider a finite system with two ports in the anomalous non-Hermitian phase and excite it. The field maps for excitation from port 1 and port 2 are shown in Fig. R6c. We see that only an edge mode is excited, and that edge mode can travel with decaying/growing amplitude depending on the way the edge is cut and the direction of propagation. The system acts as a non-reciprocal amplifier, with the energy being transmitted with amplification only from port 1 to 2, while transmission is blocked from 2 to 1. While a lot of questions remain, like the difference between the Chern and anomalous transport, the role of the edge geometry, etc, *this example demonstrates that these systems exhibit very rich physics, that are beyond the current scope of the paper and must be left to future studies.*

Figure R6: Complex Floquet band structure (real and imaginary parts) of **a, an anomalous non-reciprocal network in the presence of non-Hermitian perturbations and **b**, a Chern non-reciprocal network. Red/blue colors show localization to the top/bottom edges. We see that the edge mode is still present. **c**, Proof that the edge modes can be excited independently of the bulk modes to make a non-reciprocal amplifier. The imaginary parts acquired by the edge modes induce exponential decay along one direction and growth in the other. The skin bulk modes are not excited.**

Finally, we would like to show that the skin effect for the bulk modes is not inevitable. Consider now a PT-symmetric scenario with $l_1 = l_2 = l_3 = 0.2$ and $g_1 = g_2 = g_3 = 0.2$. The results are shown in Fig. R7. Remarkably, only the edge modes acquire a non-zero imaginary part, whereas the bulk bands remain real, meaning that the bulk skin effect does not happen. In fact, we can prove analytically that the bulk bands do not change compared to the Hermitian case. Looking at the transmission field maps, we see that the edge mode travels with exponential decay/growth, depending on the direction. The different physical properties when compared with Fig. R6 show that non-Hermitian extensions of our concept lead to a whole new world of physical properties.

Figure R7: Example of a non-Hermitian extension of anomalous Floquet network based on PT symmetry. Remarkably, the bulk bands are unchanged compared to the Hermitian case: they remain real and show no skin effect. **a**, Case of the anomalous phase. The localized edge mode acquires an imaginary part that depends on the edge type. **b**, Case of the Chern insulator. **c**, Boundary transport in a finite two-port system, showing amplification/decay of the edge mode depending on the type of edge. The system works as a non-reciprocal amplifier.

Revision: We decided not to include these extra studies to the Supplementary Information not to divert the reader from the main message and scope of the paper. We are happy though that this response letter be published alongside with the manuscript to foster future research about non-Hermitian extensions of our study.

We are grateful to Reviewer 1 for the many relevant suggestions that have led to extra studies that are now included in the revised manuscript, increasing the breadth of our work. We hope that he/she will appreciate the answers and new modifications and provide us with a positive final recommendation.

Reviewer 2

R2.1: The manuscript by Zhang et al deals with a type of anomalous nonreciprocal networks which has stronger robustness against defects than the Chern insulators. The authors establish the model of nonreciprocal topological networks, and set the reflection or the degree of nonreciprocity for one unit-cell as an indicator of different topological phases of the network. Further, the authors find that when the reflection is below a threshold or the degree of nonreciprocity for the building block is large enough, the system will support a type of anomalous edge mode that has superior robustness over the Chern one against the phase link disorder of an arbitrarily large value. The authors proved by simulations and experiments that the proposed structure can achieve good one-way wave transport.

After going through this work, I have the following concerns which requires the authors to clarify

Response: We thank the Referee for providing us with constructive and very useful remarks, which have led to several important revisions that have made our work stronger.

R2.2: 1. The first concern is about the definition of anomalous phase, phase transition and Chern phase. The similar nonreciprocal topological networks have been constructed previously, for example, in acoustics (Phys. Rev. Lett. 122, 014302, 2019, not cited in this work).

Response: This prior art is indeed a nice experimental work in acoustics, but there exist important differences with our work. First, this experiment on a network of acoustic circulators does not study the Floquet topology. There is also no mention or study of the robustness to distributed disorder. Technically speaking, absorption losses are much more significant in such acoustic networks, and size constraints limit these experiments to networks with a small number of scatterers. Both the focus, results, and physical platform are therefore very different from our paper. That being said, we are happy to include this work to our reference list.

Revision: The work has been added to our reference list (Ref. 25).

R2.3: The main discovery here is that the degree of nonreciprocity or the reflection of one unit-cell can be tailored to drive the network into two different topological phases. For example, the low reflection or high degree of nonreciprocity case is termed as an anomalous phase, with the quasi-energy band structure featured with the topological gapless edge modes in all band gaps. The high reflection or low degree of nonreciprocity case is termed as the Chern phase, with the quasi-energy band structure featured with some band gaps trivial of no topological edge modes. However, in PRL 122, 014302, 2019, the discussed case seems like to be the anomalous phase in this work, where the paper named it as the Chern phase and the network as acoustic Chern insulator, since the Chern number is calculated to be non-zero. So, I have a question that what is the topological invariant or Chern number of the bands in Fig. 1c?

Response: The Chern numbers of all bands in the left panel of Fig. 1c are rigorously zero, and the ones of the right panel are 0,-1,1,0,-1,1 from bottom to top.

However, the Chern number does not fully account for the topology of unitary operators, such as the scattering matrix in Eq. 1. As explained by Rudner¹ *et al.* for unitary evolutions, the eigenvalue (quasi-energy) spectrum being defined on a circle, it is now allowed for each (quasi-energy) band to be connected to the next one by an edge state. Because of the cyclicity of the spectrum, and because the Chern number of a band counts the number of edge states that merge into that band, it follows that the Chern numbers of each band vanish. Since all the gaps are filled by a chiral edge state, this regime is called anomalous.

Actually, the topology of unitaries, such as evolution operators or our scattering matrix, are better described by the homotopy group $\pi_3(U(N)) = \mathbb{Z}$, whose elements are the topological numbers

$$W_\psi = \frac{1}{24\pi^2} \int \text{tr}(V_\psi^{-1} dV_\psi)^3.$$

The power 3 must be understood in the language of differential forms, and the integral runs over a 3-torus, spanned by the quasi-momentum $\mathbf{k}=(k_x, k_y)$ and time t (over a time period T). Time is not explicit in scattering networks. However, the *cyclicity* of the network makes possible a direct mapping with a Floquet (i.e. T -periodic in time) evolution operator $U(t, \mathbf{k})$, such that an interpolation parameter that formally plays the role of time, can be introduced². Finally, the operator V_ψ is a periodized (in time) evolution operator. For Floquet systems, it reads as¹

$$V_\psi(t, \mathbf{k}) = U(t, \mathbf{k})e^{itH_{\text{eff}}(\mathbf{k})}$$

with

$$H_{\text{eff}}(\mathbf{k}) = \frac{i}{T} \ln_{-\psi} U(t = T, \mathbf{k}),$$

where $-\psi$ denotes the branch-cut of the logarithm. The procedure to define such an operator V_ψ and thus the invariant W_ψ for discrete-time evolutions, (i.e. when the dynamics is given by a succession of scattering events and where time therefore does not appear explicitly), like in our model, was developed in a previous detailed study² (in particular in sections V. A. and V. B.).

Importantly, the branch-cut ψ must be chosen in a spectral gap of $U(T, \mathbf{k})$, or $S(\mathbf{k})$ in our case. For this reason, W_ψ is said to be a gap invariant, and indeed directly gives the number of chiral edge states in a given quasi-energy gap ψ . In contrast, Chern numbers are band invariants. They are inferred from $H_{\text{eff}}(\mathbf{k}) = \frac{i}{T} \ln_{-\psi} U(t = T, \mathbf{k})$ and thus cannot capture the full unitary evolution.

Finally, the details for the calculation of the invariants W_ψ in oriented Kagomé graphs can be found in Delplace² *et al.* Their values for the band structures of Fig. 1c of the main text are 1,1,1,1,1 in the anomalous case and 1,0,1,1,0,1 for the Chern case.

Revision: We have added more details about the topological characterization of the system in a new section of the Methods.

R2.4: 2. The superior robustness of the anomalous phase can be well understood from the quasi-energy band structures in Fig. 2. After a further thought, it can be found that the superior robustness is specific to the link-phase variations but not to the structure of the circulators or the operation frequency as shown in the Extended Data Fig. 4a. Intuitively, the link-phase variation does not change the degree of nonreciprocity for each unit-cell in the network. Also for the AFI case, the superiority is limited by the existence of bulk bands in quasi-energy band structures, even though the case of CI is worsened due to the existence of additional band gaps.

Response and revision: The robustness to phase link disorder is exactly what the left panel of Fig. 3c proves in a fully quantitative way, and the additional measurements of Fig. 4d now prove it experimentally. Regarding the effect of the bulk bands, we agree with the Referee: in fact, their contribution to wave transport can even drop to zero at the phase rotation symmetric point (perfect circulator case), where the bands must be flat, in contrast with the Chern phase that cannot support this condition. It is truly remarkable that there exists a drastic qualitative difference between CI and AFI for such a physically relevant type of disorder.

Nevertheless, in addition to the case of phase disorder, we now demonstrate that the superior robustness of the AFI phase does apply to the other possible parametric variations in the network, namely S-parameter disorder (see R1.12, and new right panel in Fig. 3c).

R2.5: 3. From the Extended Data Fig. 7b, the experimental measurement shows an obvious field attenuation in comparison to simulation. For a much larger network in practical use, the practicability test would fail if the field attenuation arises. In my viewpoint, the wave attenuations due to the reflection/scattering and the absorption are equally important to be solved.

Response: Here, we used a substrate with a loss tangent of 10^{-4} and circulators with insertion loss of 0.2 dB, and therefore absorption has been kept to the smallest possible value allowed by current microwave technologies. Components in our cellphones have similar level of losses, and yet they work and are used ubiquitously. Our experiment with a system of 70 circulators does show some attenuation, but it is still kept to a very acceptable value (20 dB), much higher than the typical dynamic range that current communication systems can support (at least 50 dB). We do not think that applications would require a larger number of scatterers, but if this is the case, the use of active strategies, such as amplifiers or PT-symmetric gain/loss distributions as the one proposed in Fig. R7 could be a solution.

Revision: We now comment on this point when discussing the assumption of unitarity in a newly added Methods section.

R2.6: 4. I found there may be a typo in the manuscript. In the Measurement setup part, it should be "we connect ports 1 and 4 to the two VNA ports, while letting ports 2 and 3 open."

Response and revision: Thank you for detecting this glitch. This is fixed in the revised manuscript.

R2.7: Besides those concerns, I also have some suggestions for the authors: 1. For Fig. 4, it will be more interesting if the authors can introduce some link-phase disorders into the Switzerland shaped network.

Response: The Referee suggests to make extra experiments to explicitly show the resilience of the edge mode to large phase-link disorder. We totally agree, this is a great suggestion.

To address the comment, we have built five new samples with an irregular shape, and five different randomly-generated realizations of phase link disorder (see Fig. R10 below within the response to Reviewer 3). The phases fluctuate from 0 to 2π (fully random case). We decided to prototype five different random realizations instead of a single one, to rule out the possibility of a lucky disorder realization. To reduce prototyping cost, we work with slightly smaller samples than the Switzerland network. The measured field maps in the anomalous and Chern phases are also shown in Fig. R10. Regardless of the disorder realization, the anomalous edge mode is not affected, whereas the Chern one is totally destroyed. This is clear experimental evidence of the resilience of the anomalous phase even in the a fully random disorder case. These field maps are in complete agreement with the numerical predictions of the right panel in Fig. 3c of the main text, as well as numerical predictions shown in Supplementary Information Fig. S4.

Revisions: We have revised Fig. 4, which is reproduced below, to include the field maps measured for one of the new prototypes. The four other are shown in Extended Data Fig. 9. We also add relevant numerical predictions in the Supplementary Information (Fig. S4). The main text has been updated to discuss the new results.

Figure R8: *New version of the last figure of the main text, including a new panel d, showing one of our new prototypes built to demonstrate experimentally the superior robustness of the anomalous phase over the Chern one. The 2-port network has an irregular shape, and exhibits fully random phase delays implemented by varying the length of the serpentine connections between the circulators (see inset). The edge mode amplitude distribution in the anomalous band is localized to the edge and reaches port 2, whereas in the Chern band, it is completely blocked. The same behavior is observed for the four other realizations of disorder that we prototyped (see Fig. R10).*

R2.8: 2. In Fig. 2b, the authors can show the top edge mode and bottom edge mode for the three cases in Fig. 2a. I am wondering whether there exists any difference in field distributions among the edge modes for the anomalous phase, phase transition point and Chern phase.

Response and revision: We have revised Fig. 2 to plot the field distributions of all edge modes. The Chern edge mode is slightly less localized than the anomalous one, due to the higher level of reflection at the circulators.

R2.9: 3. For Extended Data Fig. 3, the authors can show the measured transmission for AFI and CI to verify the simulation result.

Response: Following the Reviewer's suggestion, we generated the same figure but with the experimental data, obtaining Fig. R9. The band structure and transmission spectra are in full agreement. Note that in the

experiment, the scattering matrices are not perfectly C_3 -symmetric, not perfectly unitary, and dispersive, leading to a slight deformation of the band structures when compared to the numerical predictions. These effects are taken into account in the experimental design (as described in Methods).

Figure R9: New supplementary figure showing the agreement between the experimentally measured transmission and the expected band structure, as suggested by Reviewer 2.

Revision: The two corresponding figures (comparing simulated and measured cases) are now included in the Supplementary Information Fig. S2 and S3.

R2.10: Overall this work strikes me as an interesting concept and a nice experimental demonstration, but both the theoretical and experimental validation in the current form seems slender to merit publication in Nature.

Response: We warmly thank the Reviewer for the great comments. We hope that the extra studies and measurements performed, following the suggestion of the Reviewer, will be appreciated, and that the Reviewer will find our responses and revisions appropriate. We believe that the paper has been significantly improved, with the addition of many new important results, and hope that the Referee will now appreciate our findings to their full potential.

Reviewer 3

R3.1: In the manuscript titled "Anomalous nonreciprocal topological networks: stronger than Chern insulators", the authors have reported on the design and realization of a time-reversal symmetry-breaking anomalous Floquet insulator (AFI). By changing the transmission coefficient, this system can be switched between the AFI phase, the Chern insulator (CI) phase, and the normal insulator phase. The authors confirm that AFI is more resistant to the phase link disorder compared to the Chern insulator phase.

I think that the system proposed by the authors, which is insensitive to disorder and defects, is an interesting physical system and has potential applications. However, the current paper lacks a theoretically rigorous derivation to demonstrate that the AFI is more robust to impurities and defects than the CI.

Response: We thank the Referee for his positive words and interest. First, let us stress that the nature of our contribution is experimental, and that experiments are, at least in physics, the ultimate level of evidence that one can provide. Second, we would like to stress that the statistical analyses of the edge mode transmission through many disorder realizations (Fig. 3c) are exact, quantitative studies performed based on the theoretical unitary network model. They are perfectly rigorous from the theoretical standpoint, and constitute a completely valid theoretical proof that the AFI phase is more robust than the CI. To clear any remaining doubt, we have built 5 new prototypes with 5 distinct random disorder realizations of fully disordered phase delays. We decided to explore five different random realizations instead of a single one, to rule out the possibility of a lucky disorder realization. The measurements, summarized in the Fig. R10 below, are in perfect agreement with our theory: the Chern edge mode does not go through, whereas the anomalous one is perfectly transmitted. All these evidences constitute a solid ground that is sufficient to establish our findings. Studies based on different approaches, such as random matrix theory, can be the subject of follow-up papers by experts of these methods.

Figure R10: Experimental evidence of the robustness of the anomalous edge mode to full phase link disorder. *a*, We built five new prototypes with randomly drawn realizations of phase link disorder (fully random case), implemented by varying the length of the serpentine paths connecting the scattering nodes. The two-port disordered networks (DN) are numbered from 1 to 5, and have an irregular external shape. *b*, Field amplitude distribution measured in the anomalous band. We can see an edge mode propagating from 1 to 2 in all cases. *c*, Field amplitude distribution in the Chern band. The edge mode never propagates to port 2 and is arrested by the fully random phase disorder, consistent with our statistical predictions.

Revision: We have made the experimental demonstration of our findings more complete by adding our extra measurements of fully-disordered phase links to the paper (revised Fig. 4). Fig. 3c also addresses other disorder types, hereby extending the superiority of the anomalous phase in the case of scattering matrix disorder.

R3.2: I would like the authors to consider the following questions related to their experiment and analysis.

1) The authors compared the AFI state with the CI state and chose the CI state which has boundary states in band gap type 1 but not in band gap type 2. While the AFI state has boundary states in both types of gaps. This is one of the main reasons why the transport properties of the AFI boundary state can resist stronger phase link disorder than CI. However, if the appropriate parameters are chosen or the system is well-designed, it is possible to make the boundary states exist in all gaps of the CI state.

Response: We respectfully disagree. If an edge state exists in each gap, the Chern number of each band vanishes, because the Chern number of a band precisely counts the net number of edge states that enter this band. So a regime where an edge state exists in all gaps has necessarily Chern numbers zero, and defines the anomalous phase.

R3.3: For this type of CI state, it should have the same robustness to phase link disorder as AFI. Therefore, the authors' evidence that AFI is more robust to phase disorder than CI is not very convincing and rigorous. The authors should better prove rigorously that AFI has a certain universal physical mechanism which makes it more robust to phase link disorder than CI.

Response: We must disagree. The type of CI state the Reviewer is referring to does not exist (see response to previous comment R3.2).

R3.4: 2) The authors report that the boundary transport properties of the AFI system are robust to the disorder of the phase φ . Is this related to the fact that φ appears as an exponent in $\exp(-i\varphi)$ in the eigenvalue equation?

Response: The fact that φ appears as an exponent in the eigenvalue equation is important to justify the application of Floquet topological classification to our system (the phase φ lives on a compact dimension, i.e. a circle), and thus to make possible the existence of an AFI, and not only that of CI.

While in the clean system, the phase φ is fixed (similarly to the Fermi energy in electronic transport), phase link disorder randomly shifts the value of φ . The boundary transport properties of an AFI are thus particularly robust to that disorder, because edge states with the same chirality are accessible at (almost) any φ randomly picked by disorder. In contrast, CI have spectral gaps with no edge state that are also selected by phase disorder.

R3.5: Does this form of eigenvalues result in more stability for impurities and defects than the ordinary form of eigenvalue equation? For other types of impurities, such as lattice distortion or the disorders on ω_{\pm} , does the system possess a similar high degree of robustness?

Response: This is a very interesting comment. Besides disorder on the phase links, one could indeed imagine another type of physically relevant disorder: on the S matrices of the elements. Considering disorder on the entire scattering matrix includes at the same time disorder on the circulator matching (reflection coefficient), non-reciprocity level, and scattering phases, which could occur due to the presence of impurities or finite geometrical tolerances in the circulator fabrication process.

We performed a new statistical study on scattering matrix disorder, and found that the anomalous phase is also much more robust than the Chern one in this case. Please read the response to comment R1.12 for details.

Revision: The main text now also includes the new statistical study demonstrating the superior resilience of the anomalous phase to scattering disorder. Fig. 3c has been revised accordingly. Two figures showing particular S-matrix disorder realizations have been added to the Supplementary Information Fig. S5 and S6.

R3.6: 3) Which class does the AFI belong to according to the classification of topological insulators?

Response: It belongs to class A, like CI. A topological classification for unitaries based on K-theory is presented by Roy³ *et al.* The classification is basically the same, up to the important difference that several invariants are now required (per symmetry-class and dimension) in contrast with the usual ten-fold way of topological insulators. This is due to the compactness of the spectrum of unitaries, in contrast to that of Hermitian operators. In our case, it translates to the use of an additional topological number to fully capture the topology of our scattering network (see R2.3 for all the details about the topological invariants).

R3.7: Which topological number is used to characterize its topological properties? How does the topological number change when changing from AFI state to CI state?

Response: The topological numbers that are used are the elements of the homotopy group $\pi_3(U(N)) = \mathbb{Z}$ that classify unitaries such as our scattering matrix. Those numbers, quoted W_ψ , are defined for each spectral gap ψ , in contrast with Chern numbers C_n that characterize the fiber bundles defined for each eigenstate Ψ_n (and thus each band n) over the Brillouin zone. A detailed discussion of the topological invariants, including definitions, is provided in response to comment R2.3.

Revision: We have added a new Methods section describing the topology of the network. We have also included the band gap map as a function of reflection and the associated gap invariants of the systems in the Supplementary Information (see R3.9).

R3.8: Can a more detailed form of S(k) be given in Equation 1 so that it can be clearly distinguished whether the system is in the CI or AFI state.

Response: The detailed formula for the matrix $S(k)$ involved in Eq. 1 is unfortunately too complex to be included in the main text, or to directly infer information on the topological phase diagram.

It is given by the following matrix product:

$$S(\mathbf{k}) = \begin{bmatrix} 0 & e^{-ik \cdot \alpha_2} & 0 \\ 0 & 0 & e^{-ik \cdot \alpha_1} \\ 1 & 0 & 0 \end{bmatrix} \begin{bmatrix} 0 & 0 & 1 \\ e^{ik \cdot \alpha_2} & 0 & 0 \\ 0 & e^{ik \cdot \alpha_1} & 0 \end{bmatrix} \begin{bmatrix} R & T & D \\ D & R & T \\ T & D & R \\ R & T & D \\ D & R & T \\ T & D & R \end{bmatrix},$$

where $R = -1 + \frac{2}{3} \cos \xi e^{i\xi} + \frac{2}{3} \cos \eta e^{i\eta}$, $T = \frac{2}{3} (e^{-i\frac{2}{3}\pi} \cos \xi e^{i\xi} + e^{i\frac{2}{3}\pi} \cos \eta e^{i\eta})$, and $D = \frac{2}{3} (e^{i\frac{2}{3}\pi} \cos \xi e^{i\xi} + e^{-i\frac{2}{3}\pi} \cos \eta e^{i\eta})$.

Revision: This expression is now provided in the Methods, Eq. 8.

R3.9: Can the authors give the topological energy gap as a function of phase links and the formula for the topological number?

Response: The formula for the topological number W_ψ is given above in R2.3. The phase links (the quasi-energy) is what plays the role of the energy in more standard topological insulators (Floquet insulators work with different spectral projectors). Therefore, the gap amplitude cannot depend on the phase link. Instead, we found that it only depends on the level of reflection R of the individual scatterers. We can therefore make the following plot of the topological gap vs. reflection:

Figure R11: Band gap map of the network. The white areas represent bulk bands. The blue areas represent band gaps with values of the homotopy invariant $W_\psi = 1$, whereas red areas correspond to band gaps with a zero value of W_ψ .

Revision: We have included the band gap map as a function of reflection and the associated gap invariants of the systems in the Supplementary Information.

R3.10: 4) How strong does the external magnetic field required to break the time-reversal symmetry?

Response: The required magnetic field is very reasonable: around 628 Oe. It is provided by tiny magnets in our experiment, located inside the circulators. We provide below the result of the finite-element full-wave simulation performed to model the circulators. Strong non-reciprocity is obtained for a bias magnetic field of only 50kA/m= 628 Oe.

Figure R12: Full wave finite element simulation demonstrating that an external magnetic field of 50kA/m is sufficient to obtain a strong non-reciprocal response of the circulators.

Revision: We now mention the value of the external magnetic field in the Methods.

We thank Referee 3 for the pertinent review, which has led to significant revisions and improvement of both our results and of their presentation. We hope that he/she will acknowledge the added value of our revisions and of the new experiment, such as to be able to express a positive recommendation for our manuscript.

References:

1. Rudner, M. S., Lindner, N. H., Berg, E. & Levin, M. Anomalous Edge States and the Bulk-Edge Correspondence for Periodically Driven Two-Dimensional Systems. *Phys. Rev. X* **3**, 031005 (2013).
2. Delplace, P., Fruchart, M. & Tauber, C. Phase rotation symmetry and the topology of oriented scattering networks. *Phys. Rev. B* **95**, 205413 (2017).
3. Roy, R. & Harper, F. Periodic table for Floquet topological insulators. *Phys. Rev. B* **96**, 155118 (2017).

Reviewer Reports on the First Revision:**Ref #1**

In the revised manuscript and response, the authors have seriously considered my comments (as well as the other referees'), and carried out additional experimental investigations of disorder. In particular, I am now very convinced that their anomalous Floquet system does indeed possess far superior robustness compared to ordinary Chern systems (from figs R1 to R4 and surrounding discussions), and that their experiment is an important milestone in this new paradigm.

Their study of non-hermicity from loss does possess interesting conclusions in their own right (R6 and R7), particularly pertaining to the nonhermitian skin effect, but I agree that those investigations are not directly relevant to the merits of this experiment and can appear as interesting future work.

Overall, I hereby strongly recommend this work for publication in Nature.

Ref #2

I have went through the revised manuscript very carefully. I understand that the authors have made great efforts in providing more simulation and experiment results to support the key conclusions that they draw. However, I still have the following concerns which requires the authors to clarify

1.For the reply of R2.2, I have found another interesting work of three dimensional Floquet acoustic networks, where the loss is much less and the size constraint is released to allow for a much larger array (Physical Review Research 1(3), 033149, 2019). So is it a type of anomalous phase that is stronger than the Chern insulator? A detailed comparison between this work and previous ones is necessary and can make this Nature-candidate work of less debate.

2.For the demonstration of superior robustness of AFI against parameter variations in the S-

parameter disorder, I find that the disorder is still limited to the constraint parameter regions of AFI and CI (different triangle regions of η and ξ), respectively. Well, if the introduced disorder is large enough, let's say the parameters of AFI unfortunately go to CI region, the topological phase will not be that robust due to the existence of band gaps or bulk bands, as demonstrated by the authors.

3. Basically, the configuration proposed in this work is simple and not difficult to be discovered (also quite surprising to me). The graphene network consisting of giant nonreciprocal circulators can have strong topological transport, which is acceptable. So my concern is that why no one has ever discovered AFI previously or maybe someone has discovered the anomalous phase in another form, either theoretically or experimentally, or in other wave systems. I hope the authors can give me some unequivocal evidences that show its complete novelty and the stark differences from previously discovered topological phases.

If those questions can be well tackled, I can recommend its publication in Nature.

Ref #3

Report from Reviewer 3

The authors have made a large improvement in the revised manuscript. The issues raised by the reviewers, such as the topological classification of the system, the topological number, and the effect of different kinds of disorders on the system, have been carefully addressed.

In the revised manuscript, the authors experimentally implemented the anomalous Floquet phase. As pointed out by the reviewers I and II, Floquet topological insulators have been studied in several systems and are not new concepts nor new state of matters. The authors focus on the transmission properties of the boundary states of anomalous Floquet phase. As shown in Figure 2a and Figure R9, the authors show that the bulk states of the anomalous Floquet phase have narrower bandwidths and boundary states are present in all gaps, while the bulk states of the Chern phase have wider bandwidths and there are gaps without boundary states. These properties make the transmission properties of the boundary of the anomalous Floquet phase to be more robust to phase links disorder and scattering matrix disorder.

For a manuscript submitted to Nature, the lack of a rigorous mathematical proof to reveal whether there is a universal physical mechanism to support the authors' findings is an imperfectness. But the authors have added new experiments with statistical analyses to support their findings. The results of these experiments may stimulate theorists to investigate the physical mechanisms behind them. However, the authors still need to address the following issues to make their experimental conclusions more convinced.

(1) The Chern phase chosen by the authors (Fig. 2a, right) has two gaps without boundary states. This kind of gap has a disadvantage on the robustness of the boundary transmissions. Therefore, it is prone to conclude that the anomalous Floquet phase is superior to the Chern phase. The authors need to consider the case where the Chern phase only has a single gap without boundary states. Such a Chern phase exists and its robustness to disorders will be enhanced. Therefore, the authors need to consider whether the advantage of anomalous Floquet phase still exists in such a case, and if it does, whether it has a very significant advantage.

The more general case is that for a system with n energy bands and n gaps, the anomalous Floquet phase may have boundary states in all n gaps, while the Chern phase may have boundary states in $n-m$ gaps and no boundary states in m gaps. When m/n is large (e.g., $m=2$ and $n=5$ for the system given by the authors), the transmission properties of the anomalous Floquet phase are superior to those of the Chern phase. But if m is small and n is large, i.e., m/n is small, does the robustness of the boundary states to disorders for the anomalous Floquet phase and the Chern phase tend to be the same?

(2) For the Chern phase, if there are multiple boundary states in the gap, the more boundary states increase the transmission capacity on the boundary. Does it make the robustness to disorders of the Chern phase and the anomalous Floquet phase to be close?

(3) Tuning the parameters of the Chern phase to make the bandwidth of the bulk state smaller, or to make the gap with the boundary state larger and the gap without the boundary state smaller, intuitively thinking, is also able to improve the robustness of boundary state to disorders for the Chern phase.

To summaries, to experimentally demonstrate that the anomalous Floquet phase is more robust to disorder than the Chern phase, it is necessary to take into account not only the effects of multiple types of disorders, but also need to compare to various types of band structures as mentioned above. In this way the results of the experiment can be made more comprehensive and more convincing. Based on the above considerations, until the above issues are resolved, I am reluctant to recommend the article to be published in the high standard Nature journal.

Author Rebuttals to First Revision:

Reviewer 1

R1.1: In the revised manuscript and response, the authors have seriously considered my comments (as well as the other referees'), and carried out additional experimental investigations of disorder. In particular, I am now very convinced that their anomalous Floquet system does indeed possess far superior robustness compared to ordinary Chern systems (from figs R1 to R4 and surrounding discussions), and that their experiment is an important milestone in this new paradigm.

Their study of non-hermicity from loss does possess interesting conclusions in their own right (R6 and R7), particularly pertaining to the non-hermitian skin effect, but I agree that those investigations are not directly relevant to the merits of this experiment and can appear as interesting future work.

Overall, I hereby strongly recommend this work for publication in Nature.

Response: We thank the Reviewer for his/her careful study of our revised manuscript and response, and are happy to see his/her strong positive recommendation for publication.

Reviewer 2

R2.1: I have went through the revised manuscript very carefully. I understand that the authors have made great efforts in providing more simulation and experiment results to support the key conclusions that they draw. However, I still have the following concerns which requires the authors to clarify

Response: We thank the Referee for carefully going through our revisions. We are happy to provide some clarifications about the new comments made by the Referee.

R2.2: 1. For the reply of R2.2, I have found another interesting work of three dimensional Floquet acoustic networks, where the loss is much less and the size constraint is released to allow for a much larger array (Physical Review Research 1(3), 033149, 2019). So is it a type of anomalous phase that is stronger than the Chern insulator? A detailed comparison between this work and previous ones is necessary and can make this Nature-candidate work of less debate.

Response: This *reciprocal* structure cannot implement the *non-reciprocal* physics that we discuss. The reason is that such networks mimic time-Floquet topological insulators by emulating time using a third spatial dimension. Contrary to real time, that never allows a return of waves in the past, in reciprocal systems waves traveling along a negative spatial dimension are allowed. This is the largest and most crucial difference between our work and the one mentioned by the Referee. Other than that, this study of a 3D acoustic system, albeit very interesting, did not consider any measurement of transmission, and did not consider studying the effect of disorder, even for this different reciprocal scenario.

Our 2D microwave non-reciprocal system is the first experiment reporting, studying and evidencing the differences in transport properties between the anomalous and Chern phases. As acknowledged already by the other Referees, there is no debate about this.

R2.3: For the demonstration of superior robustness of AFI against parameter variations in the S-parameter disorder, I find that the disorder is still limited to the constraint parameter regions of AFI and CI (different triangle regions of η and ξ), respectively. Well, if the introduced disorder is large enough, let's say the parameters of AFI unfortunately go to CI region, the topological phase will not be that robust due to the existence of band gaps or bulk bands, as demonstrated by the authors.

Response: Obviously, to compare the two phases, we need to stay within each phase. Studying a mixture of Chern and anomalous does not make any sense when one wants to compare their robustness.

R2.4: Basically, the configuration proposed in this work is simple and not difficult to be discovered (also quite surprising to me). The graphene network consisting of giant nonreciprocal circulators can have strong topological transport, which is acceptable. So my concern is that why no one has ever discovered AFI previously or maybe someone has discovered the anomalous phase in another form, either theoretically or experimentally, or in other wave systems. I hope the authors can give me some unequivocal evidences that show its complete novelty and the stark differences from previously discovered topological phases.

Response: The fact that our platform is relatively simple is not a scientifically valid argument to argue about novelty. Novelty can only be questioned by providing concrete examples of relevant previous literature. Accordingly, we have considered all the works that we know about, as well as the ones previously cited by the Referee, and discussed the important differences with ours. Besides, its relative simplicity makes its impact for potential technological applications especially relevant.

The stark differences with previous works are the following. We, for the first time, studied the *transport* properties of truly *non-reciprocal* anomalous and Chern Floquet edge states, and *found a markedly different behavior*, which we explained using intuitive physical arguments, and validated both with statistical numerical studies and experiments. Our platform is compatible with current planar microwave technologies, which will be an advantage for future explorations of technological applications of our findings, for example in 5G communication systems.

R2.5: If those questions can be well tackled, I can recommend its publication in Nature.

Response: We believe to have made the novelty of our work very clear in our responses above. We thank Reviewer 2 for the careful review.

Reviewer 3

R3.1: The authors have made a large improvement in the revised manuscript. The issues raised by the reviewers, such as the topological classification of the system, the topological number, and the effect of different kinds of disorders on the system, have been carefully addressed.

In the revised manuscript, the authors experimentally implemented the anomalous Floquet phase. As pointed out by the reviewers I and II, Floquet topological insulators have been studied in several systems and are not new concepts nor new state of matters. The authors focus on the transmission properties of the boundary states of anomalous Floquet phase. As shown in Figure 2a and Figure R9, the authors show that the bulk states of the anomalous Floquet phase have narrower bandwidths and boundary states are present in all gaps, while the bulk states of the Chern phase have wider bandwidths and there are gaps without boundary states. These properties make the transmission properties of the boundary of the anomalous Floquet phase to be more robust to phase links disorder and scattering matrix disorder.

For a manuscript submitted to Nature, the lack of a rigorous mathematical proof to reveal whether there is a universal physical mechanism to support the authors' findings is an imperfectness. But the authors have added new experiments with statistical analyses to support their findings. The results of these experiments may stimulate theorists to investigate the physical mechanisms behind them. However, the authors still need to address the following issues to make their experimental conclusions more convinced.

Response: We thank the Reviewer for accepting our previous responses, and we are happy to provide detailed answers to his/her remaining comments. We have carefully studied them, and performed extra statistical studies that are presented below.

R3.2: (1) The Chern phase chosen by the authors (Fig. 2a, right) has two gaps without boundary states. This kind of gap has a disadvantage on the robustness of the boundary transmissions. Therefore, it is prone to conclude that the anomalous Floquet phase is superior to the Chern phase. The authors need to consider the case where the Chern phase only has a single gap without boundary states. Such a Chern phase exists and its

robustness to disorders will be enhanced. Therefore, the authors need to consider whether the advantage of anomalous Floquet phase still exists in such a case, and if it does, whether it has a very significant advantage.

Response: We have studied the case mentioned by the Referee, namely a Chern phase with a single trivial gap, and found that still, it is far from being as robust as the transmission through the anomalous edge mode.

In our network, such a phase cannot be directly obtained, because the matrix $S(\mathbf{k})$ satisfies the relation $PS(\mathbf{k})P^\dagger = -S(\mathbf{k})$, with $P = \text{diag}(\mathbb{1}_3, -\mathbb{1}_3)$, which implies that the bulk band structures have π -translation symmetry, forcing the gaps to close and open by pairs. The π -translation symmetry can be broken if we introduce an extra unitary 2-port reciprocal scatterer in the middle of the connecting links. By setting the reflection of this extra scatterer to 0, we recover the previous network. However, playing with the reflection level of this extra scatterer allows us to extend our parameter space and reach the cases mentioned by the Referee.

Noting M the number of trivial band gaps, we generate the cases $M=1$ and $M=3$. The phase $M=0$ is the anomalous phase, and $M=2$ is the Chern phase already reported in the paper. Band structures with edge modes are represented in Fig. R1a, for the cases $M=0, 1$, and 3 . Next, we perform our statistical transmission analysis on a finite network with 1000 different realizations of phase link disorder, with range up to 2π . Panel b shows the results. **Clearly, even the $M=1$ case, despite having a small trivial band gap (of size $\pi/6$), is largely affected by the disorder.** $M=3$ is worse than the case $M=2$ presented in the main text, as expected.

Revision: We added Figure R1 to the Supplementary Information as Fig. S9.

Figure R13: Effect of the number of trivial band gaps (noted M) on the robustness of the edge mode transmission to phase link disorder. **a, Ribbon band structures. We compare the case of the anomalous phase (left), in which all band gaps exhibit edge modes, to the case mentioned by the Reviewer ($M=1$, center) of a Chern phase with a single trivial gap, which is the most favorable scenario for Chern. For a complete comparison, we also take the case $M=3$ (right). The case $M=2$ is already studied in the main text. **b**, Transmission statistics (average, Q1 and Q3) for the three cases for 1000 realizations of random disorder.**

R3.3: The more general case is that for a system with n energy bands and n gaps, the anomalous Floquet phase may have boundary states in all n gaps, while the Chern phase may have boundary states in $n-m$ gaps and no boundary states in m gaps. When m/n is large (e.g., $m=2$ and $n=5$ for the system given by the authors), the transmission properties of the anomalous Floquet phase are superior to those of the Chern phase. But if m is small and n is large, i.e., m/n is small, does the robustness of the boundary states to disorders for the anomalous Floquet phase and the Chern phase tend to be the same?

Response: Our extra study, summarized in Fig. R1, shows that by working with Chern phases with less trivial band gaps, we cannot reach the same level of robustness than the anomalous phase.

Here, the number of gaps is $N=6$, and in our response to comment R3.2, we have considered the cases $M=0$ (anomalous), $M=1$ (the best Chern), and $M=3$ (a less robust Chern phase). The case $M=2$, already previously reported (see Fig. 3c in the main text), falls between the $M=1$ and $M=3$ curves, and is not reproduced to avoid burdening the figure. Our quantitative data thus confirms the general intuition of the Referee that less trivial gaps improve the Chern transmission, however even with only one gap, the Chern phase remains much less robust.

Revision: We added Figure R1 to the Supplementary Information as Fig. S9.

R3.4: 2) (2) For the Chern phase, if there are multiple boundary states in the gap, the more boundary states increase the transmission capacity on the boundary. Does it make the robustness to disorders of the Chern phase and the anomalous Floquet phase to be close?

Response: If we increase the Chern number C , we have more edge modes, however it does not improve the robustness of the Chern phase, as we prove below.

Figure R14: Effect of the number of Chern edge modes on the transmission robustness. a, Band structures. We compare three different Chern phases with one (left), two (center) and three edge modes (right) in the non-trivial gaps. **b, Transmission averages for 1000 realizations of random phase disorder.** As for transmission between two antennas in a multimode environment, the input power has to split over the various transmission channels which interfere at the output. Disorder creates random interferences between the different channels at the output, which is statistically detrimental to transmission.

To increase the Chern numbers, we stacked several networks with Chern phases with $C = 1$ together and coupled adjacent layers weakly with unitary directional couplers. We considered the three cases shown in Fig. R2a, whose band structures are characterized by Chern numbers with absolute values of 1, 2, and 3, respectively (the Chern numbers for the flat bands at quasi-energies 0 and π remaining 0 here). The Chern gaps have, consequently, 1, 2 and 3 edge modes per edge (the band structure is for a ribbon, as Fig. 2a in the main text, with edge modes at both top, in red, and bottom, in blue). Next, we consider the transmission averaged over 1000 realizations of phase link disorder for the three cases (panel b). We observe that an increase of the number of edge modes does not improve the transmission. This is actually expected: the input power has to split over the various available transmission channels. Each transmission channel will contribute to transport and undergo a

different phase shift when disorder is imparted, before interfering at the output port. This multi-mode interference is statistically detrimental to power transmission.

Revision: We added Figure R2 to the Supplementary Information as Fig. S10.

R3.5: (3) Tuning the parameters of the Chern phase to make the bandwidth of the bulk state smaller, or to make the gap with the boundary state larger and the gap without the boundary state smaller, intuitively thinking, is also able to improve the robustness of boundary state to disorders for the Chern phase.

Response: As intuited by the Referee, in the fully-random (2π strength) phase-link disorder, the value of the transmission depends on the ratio in amplitude between the bandwidths of all the bulk bands, the trivial gaps, and the gaps hosting chiral edge modes. In particular, by reducing the band widths, one increases the transmission, even for the Chern phase.

Note that this mechanism is nonetheless more efficient for the anomalous phase than for the Chern phase. Indeed, the anomalous phases are related by a continuous deformation (that does not close a gap) to a phase-rotation symmetric point, where all the bulk bands are flat¹. Owing to the topological nature of the Chern numbers and of the W_ψ indices (discussed during the previous referring round), the Chern phases in scattering networks cannot be continuously deformed to such a special point, otherwise the Chern numbers would be zero. Therefore, their band widths have a minimal finite value that always reduces the transmission compared to the contribution of an edge mode. This favors the anomalous phases that can reach a perfect $T = 1$ transmission even in the fully-random phase link disorder, by tuning the scattering parameters at the green points in Fig. R3a.

Owing to this impossibility to flatten all the bulk bands, and of course to the existence of trivial gaps, it is thus clear that the optimized Chern phase - i.e. by fine tuning the scattering parameters to maximize the topological gaps and to minimize at the same time both the trivial gaps and the band widths- cannot reach the perfect transmission $T = 1$ in the fully-random phase-link disorder configuration.

This said, there is of course nothing that prevents a priori this optimized Chern phase to have a higher transmission than the worse fine-tuned 'anti-optimized' anomalous phase (i.e. with band widths as large as possible). The question is then: is this comparison, between the best optimized Chern phase and the worse 'anti-optimized' anomalous phase, representative of the average Chern and anomalous cases?

To answer this question, we have computed the average transmission for a fully-random phase-link disorder at each point of our phase diagram, i.e. for all possible tunings of the band structure (Fig. R3b). We find that the anomalous phases have a typical transmission much higher than the Chern ones, and that the average transmission (over scattering parameters of the phase diagram) of the anomalous regime is also much higher than that of the Chern phase. We also find that there are very small regions where we can pick parameters so that the transmission of the anomalous phase is smaller than the highest possible transmission of the Chern phases. However, such regions only appear close to the transition lines where the gaps are small, and both the optimized Chern and anti-optimized anomalous systems collapse to a nearly insulating phase. Elsewhere, when the gaps

¹ P. Delplace, "Topological chiral modes in random scattering networks", SciPost Physics 8, 5, pp. 081 (2020).

are well resolved, the transmission in the anomalous phase is close to 1 while the transmission in the Chern phase is close to 0.

Revision: This discussion and Figure R3 have been added to the Supplementary Information as Fig. S11.

Figure R15: Average transmission in the fully disordered phase-link case for all possible networks in the parameter space. a, Reminder of the topological phase diagram for all possible S-matrix, parametrized by ξ and η . b, Corresponding averaged transmission in the fully-disordered case. The average transmission in the Chern phase is much lower than the average transmission in the anomalous phase.

R3.6: To summarize, to experimentally demonstrate that the anomalous Floquet phase is more robust to disorder than the Chern phase, it is necessary to take into account not only the effects of multiple types of disorders, but also need to compare to various types of band structures as mentioned above. In this way the results of the experiment can be made more comprehensive and more convincing. Based on the above considerations, until the above issues are resolved, I am reluctant to recommend the article to be published in the high standard Nature journal.

Response: Based on quantitative extra studies, we have fully resolved all the extra scenarios mentioned by the Referee. We take this opportunity to thank the Referee for all the interesting comments that led to an improved version of our work (3 additional figures in the Supplementary Information).

Reviewer Reports on the Second Revision:

Ref #1

N/A

Ref #2

After going through the reply carefully, I find the novelty has been clearly pointed out and the technic aspects are convincing. I have no more questions on this work, and recommend its acceptance for publication.

Ref #3

I have studied the resubmitted version of the manuscript by Zhang and co-authors. In the present version, the authors have demonstrated through simulation and experimentation that the anomalous Floquet system has superior edge transmission properties than Chern state for a wide range of impurities and a variety of configurations of Chern insulator energy band configurations. Their findings show that the anomalous Floquet system has enormous possibilities for applications in low dissipation transmission systems. It can also contribute to the study of the relevant theory.

Since the authors have clarified all the comments well, I now recommend this work for publication in Nature.

Author Rebuttals to Second Revision:**Reviewer 2**

R2.1: After going through the reply carefully, I find the novelty has been clearly pointed out and the technical aspects are convincing. I have no more questions on this work, and recommend its acceptance for publication.

Response: We thank the Referee for his/her positive recommendation.

Reviewer 3

R3.1: I have studied the resubmitted version of the manuscript by Zhang and co-authors. In the present version, the authors have demonstrated through simulation and experimentation that the anomalous Floquet system has superior edge transmission properties than Chern state for a wide range of impurities and a variety of configurations of Chern insulator energy band configurations. Their findings show that the anomalous Floquet system has enormous possibilities for applications in low dissipation transmission systems. It can also contribute to the study of the relevant theory.

Since the authors have clarified all the comments well, I now recommend this work for publication in Nature.

Response: We thank the Referee for his/her positive recommendation.